# Dynamics of cooperative excavation in ant and robot collectives

**S Ganga Prasath[1†], Souvik Mandal[2,3†], Fabio Giardina[1†], Jordan Kennedy[1], Venkatesh N Murthy[2,3], L Mahadevan[1,3,4,5]\***

[1]School of Engineering and Applied Sciences, Harvard University, Cambridge, United States; [2]Department of Molecular and Cellular Biology, Harvard University, Cambridge, United States; [3]Center for Brain Science, Harvard University, Cambridge, United States; [4]Department of Physics, Harvard University, Cambridge, United States; [5]Department of Organismic and Evolutionary Biology, Harvard University, Cambridge, United States

**Abstract** The solution of complex problems by the collective action of simple agents in both biologically evolved and synthetically engineered systems involves cooperative action. Understanding the resulting emergent solutions requires integrating across the organismal behavior of many individuals. Here, we investigate an ecologically relevant collective task in black carpenter ants *Camponotus pennsylvanicus*: excavation of a soft, erodible confining corral. These ants show a transition from individual exploratory excavation at random locations to spatially localized collective exploitative excavation and escape from the corral. Agent-based simulations and a minimal continuum theory that coarse-grains over individual actions and considers their integrated influence on the environment leads to the emergence of an effective phase space of behaviors, characterized in terms of excavation strength and cooperation intensity. To test the theory over the range of both observed and predicted behaviors, we use custom-built robots (RAnts) that respond to stimuli to characterize the phase space of emergence (and failure) of cooperative excavation. Tuning the amount of cooperation between RAnts, allows us to vary the efficiency of excavation and synthetically generate the entire range of macroscopic phases predicted by our theory. Overall, our approach shows how the cooperative completion of tasks can arise from simple rules that involve the interaction of agents with a dynamically changing environment that serves as both an enabler and a modulator of behavior.

## Editor's evaluation

This manuscript presents a quantitative study of how ants collaborate to excavate their escape from a confining barrier. The authors provide a compelling understanding of the main mechanisms driving the excavation process. They show how cooperative escape behavior arises from a non-trivial combination of movement, interaction with the substrate, and communication between individuals. The findings are supported by extensive evidence from experimental data, numerical simulations, theoretical modeling, and robotic implementation. This is an important paper that will be of interest to a broad group of researchers working on decision-making and collective behavior in living systems.

## Introduction

Collective behavior is seen in organisms across many length scales, from the microscopic to the macroscopic (*Nowak, 2006*; *Camazine et al., 2020*; *Gordon, 1999*; *Seeley, 2009*; *Couzin and Krause, 2003*). These behaviours are often functional and serve as solutions to problems associated with tasks

**\*For correspondence:**
lmahadev@g.harvard.edu

[†]These authors contributed equally to this work

**Competing interest:** The authors declare that no competing interests exist.

that cannot be solved efficiently at the individual level and range from brood care to foraging for food, protection from enemies and predation of prey, building complex architectures etc (*Feinerman et al., 2018*; *Ocko and Mahadevan, 2014*; *Hölldobler and Wilson, 2009*; *Peleg et al., 2018*; *Rasse and Deneubourg, 2001*). Since collective behavior involves multiple individuals, this necessarily involves some form of communication and/or cooperation that takes different forms across scales - from quorum sensing in unicellular bacterium and slime molds, to the waggle dance in bees, and various forms of physical signal propagation in animal societies and human organizations (*Rasse and Deneubourg, 2001*; *Alcock, 2001*; *Pennisi, 2009*; *Nowak, 2006*; *Elster, 1998*; *Couzin and Krause, 2003*).

The importance of environmental signals is particularly clearly seen in examples of collective task execution in social insects that have a long history of documented cooperative behavior (*Hölldobler et al., 1990*; *Gordon, 1999*; *Perna and Theraulaz, 2017*; *Mikheyev and Tschinkel, 2004*). Super-organisms made of individuals respond to local stimuli with stereotypical actions that leave their 'mark' on the environment, creating a spatio-temporal memory, commonly known as stigmergy (*Hölldobler and Wilson, 2009*). While stigmergy is usually associated with scalar pheromone fields, a broader definition might include the use of signaling via chemical, mechanical and hydrodynamic means (*Buhl et al., 2005*; *Mikheyev and Tschinkel, 2004*), as has been quantified in recent studies of bees (*Ocko and Mahadevan, 2014*; *Peleg et al., 2018*). To understand how collective task execution arises, we need to understand how individuals switch from local uncoordinated behavior to collective cooperation that translates to successful task execution in different social systems. From a biological perspective, this naturally involves understanding the neural circuits, physiology and ethology of an individual. A complementary perspective at the level of the collective is that of characterizing a 'crude view of the whole', which entails the quest for a small set of rules that are sufficient for task completion, along with the range of possible solutions that arise from these rules that might be tested experimentally. And finally, given the ability to engineer minimally responsive biomimetic agents such as robots (*Rahwan et al., 2019*), a question that suggests itself is that of the synthesis of effective behaviors using these agents. This allows us to explore regions of phase space that are hard to explore with social insects, and also to learn about the robustness of these behaviors using imperfect agents in uncertain and noisy physical environments, before looking for them in-vivo.

Here we use an ecologically relevant task in carpenter ants *Camponotus Pennsylvanicus*: excavation and tunneling, to quantify the dynamics of successful task execution by tracking individual ants, create a quantitative framework that takes the form of mathematical models for agent behavior, and finally synthesize the behavior using robots that can sense and act. Our work complements and builds on earlier studies on excavation (*Buhl et al., 2005*; *Tschinkel, 2004*; *Deneubourg and Franks, 1995*; *Deneubourg et al., 2002*) in social insects that looked at the effects of population size and the role of cooperation on the efficiency of digging, while developing 1-dimensional models to understand the excavation process. We go beyond these studies by (i) quantifying the collective behavior of ants by tracking them in space-time, following the dynamics of their interaction, and the process of excavation of the confining substrate, (ii) developing a theoretical framework that couples the change in ant density, substrate density and the rate of antennation in space and time to capture the collective execution of the task in terms of a few non-dimensional parameters that define the range of behaviors of the agents, (iii) synthesizing and recreating this collective task using custom-built robots that can respond to each other and the environment . An important outcome of our study is a phase diagram that shows the emergence of different collective behaviors associated with task completion as a function of just two dimensionless parameters that characterize the local rules underlying individual behavior and the nature of communication between agents such as ants and robots.

## Materials and methods
### Excavation in carpenter ants
We start with ants drawn from a mature colony of *C. Pennsylvanicus* that consist of a queen, the sole egg layer, and workers from three morphologically different castes - major, median, and minor (*Hansen and Klotz, 2005*). Although all ants perform different tasks like foraging, nest-keeping, and brood care to varied degrees, during excavation, major ants, equipped with their large mandibles, generally take the lead role, while median and minor ants transport the debris out of the nest. Ants communicate primarily through their antennae by using them to sense pheromones released by other

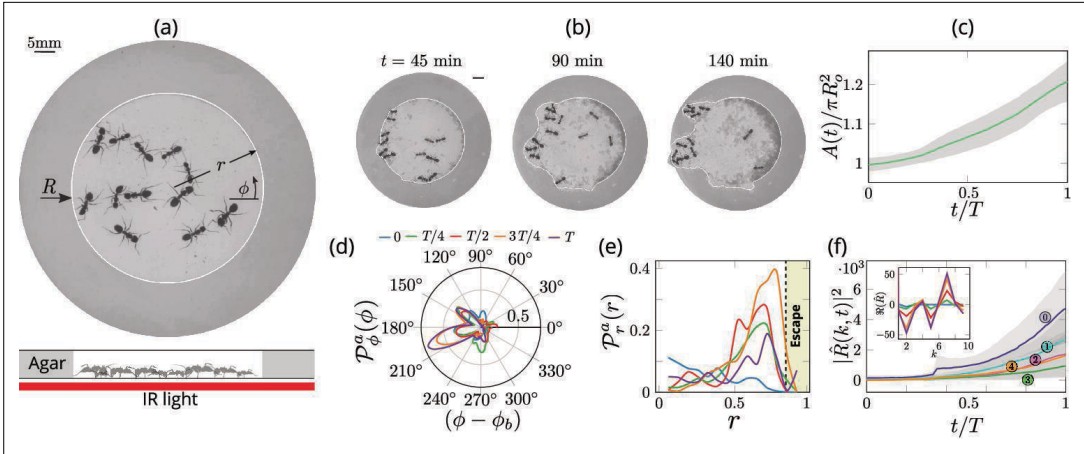

**Figure 1.** Collective dynamics of ant excavation. (**a**) Colony members of the black carpenter ant *Camponotus pennsylvanicus* are confined to a porous boundary made out of Agarose. The boundary is represented by its radius $R(\phi, t)$ ($\phi$ - polar angle, $t$ - time). Bottom part shows the side-view schematic of the experimental set-up with the boundary made of agarose and background IR light source used to image the ants in the dark. (**b**) Temporal progression of excavation experiments as 12 ants cooperatively tunnel through the agarose confinement. The white line is the tracked location of the inner wall which grows in size as the excavation progresses. (**c**) Confinement area $A(t)$ as a function of time (scaled by time to excavate out of the corral $T$), normalized by initial circular confinement with radius $R_o$. (**d**) Evolution of the orientation distribution of the ant density, $\mathcal{P}_\phi^a(\phi, t)$ obtained by averaging along the radial direction. Ants start from an initially isotropic state and localize at an angle $\phi_b$ along the boundary. $T$ here is the excavation time. (**e**) Dynamics of the radial distribution of ant density $\mathcal{P}_r^a(r, t)$ as a function of radial distance, $r$ obtained by averaging a sector of $\pi/6$ around the excavation site. We see that the ant density front propagates through the corral. The density is plotted for the same times as in (**f**) Evolution of the power spectrum $|\hat{R}(k, t)|^2$ of first five Fourier modes capturing the number of tunnels formed during excavation $R(\phi, t) = \sum_k \hat{R}(k, t) e^{ik\phi}$. Inset shows the real part of the Fourier coefficient, $\Re(\hat{R})$ at different time instants indicating that many modes are present in the boundary shape.

ants and by touching other ants to identify their caste. It is this inter-organismal information exchange that enables the collective solution of complex tasks.

Our experiments consist of a dozen worker ants from the same colony that are anesthetized (using $CO_2$) and then brought into a confining ring-like corral made out of agarose (height 10mm, inner radius 35mm and outer radius 55mm) flanked above and below by two hard plastic sheets. To mimic their natural environment in a nest, we eliminated visible light and used infrared light to monitor the ants using video (see *Figure 1(a)*). We performed 4 experiments with a collective of 12 majors ants and 3 sets of experiments with a mixture of 4 major, 4 media and 4 minor ants. Once we introduce $O_2$ into the corral, the ants regain activity but stay still for a while before moving. They first exhibit wall-following until one or more of the ants initiates an exploratory excavation at a random location along the corral (ref *Figure 2*). After an initial exploratory phase the ants switch to an exploitative strategy in which they excavate a tunnel at a specific location and eventually break through the corral (see *Video 1* and the sequence in *Figure 1(b)*). In contrast with the behavior of the 12 ant collective, when a single Major ant is introduced into the arena, the ant is unable to excavate through the agar barrier (see *Video 1*).

We can quantify this transition from rotationally isotropic exploration to localized excavation by considering both the behavior of individual ants or their effective density $\varrho_a(r, \phi, t)$ as a function of the polar coordinates $(r, \phi)$ and time $t$. We choose to use an effective coarse-grained density for two reasons: it is a more natural variable in the limit of large populations that vary in space and time, and is also amenable to building effective theories with fewer parameters that are easier to analyze and thus also compare to experiments. The ant density is obtained by averaging the position of the ants over a time window larger than the time taken for them to perform one task cycle , that starts with excavation at the boundary and ends with dropping debris in the interior of the corral (see Appendix 1 for further details). Over time, we see that the ant density becomes localized at a particular angle and location along the corral; here large-scale excavation eventually leads to excavation and escape from

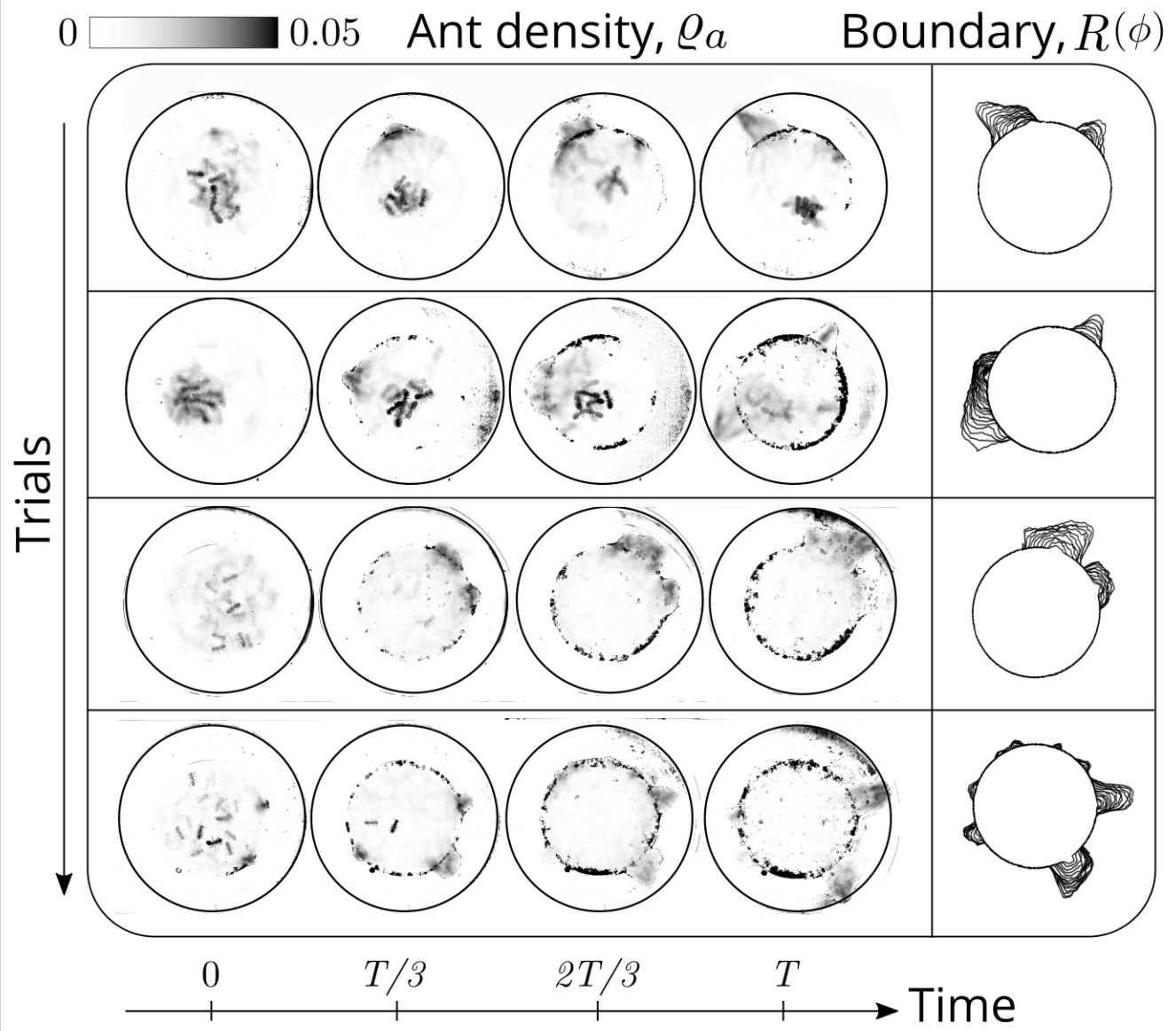

**Figure 2.** Evolution of the ant density field, $\varrho_a(\mathbf{x}, t)$ (in units of #/mm²) as the tunneling progresses for experiments with 12 major ants. The density field is obtained by averaging the ant locations over 250 s during the tunneling process. In the second columns is the evolution of the boundary shape, $R(\phi)$ as a function of time where we see multiple excavation sites being explored before one of them succeeds. The darker spots in the image are the debris that the ants deposit as they excavate the boundary.

the corral (see **Figure 2** and **Appendix 1—figure 1** for the coarse-grained spatio-temporal evolution of the ant density, obtained by this averaging procedure). Simultaneously, collective excavation leads to an increase of the volume of excavated material, as shown in **Figure 1(c)** (see also **Toffin et al., 2009**). By averaging the ant density over radial positions, in **Figure 1(d)** we show the orientation distribution of the ant density $\mathcal{P}_\phi^a(\phi, t) = \int \varrho_a(r, \phi, t)dr$ is initially isotropic, and gradually starts to localize at a particular (arbitrary) value of the angle as time increases.

Averaging the density over the localized region, in **Figure 1(e)** we show the radial distribution of the ant density $\mathcal{P}_r^a(r, t) = \int \varrho_a(r, \phi, t)S(\phi)d\phi$ (where $S(\phi)$ is a smoothing kernel localized around the excavation site) starts out by being initially uniform, and gradually propagates radially outwards as time increases. Consistent with localization and concomitant excavation (**Figure 1(f)** inset, **Appendix 1—figure 2(c)**), we see that the multiple azimuthal Fourier modes compete with each other initially before an elliptic mode (corresponding to a strongly localized state) is amplified as excavation progresses (shown in **Figure 1(f)**, **Appendix 1— figure 2(b)**). All together, our quantitative observations show that an initially isotropic and homogeneous

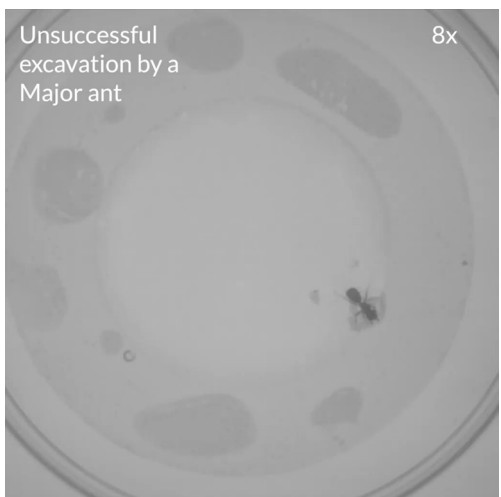

**Video 1.** Ant experiments. (*i*) Single ant: We confined 1 ant (major, media and minor individually) and capture their dynamics to see if they are capable of tunneling on their own; (*ii*) Multiple castes assemblage: We confined 12 ants, 4 for each of major, minor and media castes, and capture the dynamics of excavation as they tunnel through the boundary; (*iii*) Major ant collective excavation: We confined 12 major ants and capture the dynamics of excavation as they tunnel through the boundary.2.

https://elifesciences.org/articles/79638/figures#video1

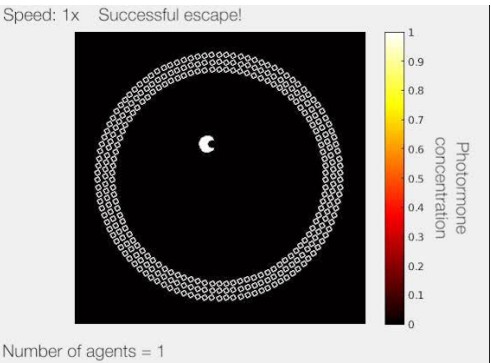

**Video 2.** Dynamics of excavation from agent-based simulation for different number of agents ($n = 1, 5, 10, 22, 100$) in the corral for parameters in tab *Appendix 2—table 1* We see successful escape as well as trapped dynamics as highlighted in Figures 3 and 4.
https://elifesciences.org/articles/79638/figures#video2

distribution of ants in the corral induces exploration of multiple potential tunneling paths that transitions into the exploitative excavation of one specific location that eventually leads to an excavation route.

## Model of cooperative excavation

In order to understand the dynamics of this cooperative excavation we first model the ants using discrete agents that mimic the microscopic behaviors of ants before turning to a coarse-grained field theoretic model for the evolution of the ant, pheromone and substrate density in space and time. In the agent-based model each ant is represented as a circular disk of radius $a$ with center position $\mathbf{r}_j(t)$ and orientation $\hat{\mathbf{p}}_j(t)$ where $j = 1 \cdots n$, $n$ being the number of ants in the domain (see *Figure 3(a)*). We approximate the confining corral in the experiments using discrete boundary elements which the agents can pick and place in the interior of the domain (see *Figure 3(b)*). Initially, a random collection of agents engages in exploration within the corral in the absence of external gradients, consistent with observations (*Trible et al., 2017*) but their motion is rectified either by the presence of pheromone gradients or reinforcing antennating signals (*Hölldobler et al., 1990*; *Reinhard and Srinivasan, 2009*; *Waters and Bassler, 2005*; *Gordon et al., 1993*; *Hillen and Painter, 2009*; *Toffin et al., 2009*). Antennation involves information moving with the ants while pheromone gradients leads to information being laid down in the fixed environment. However, when ants move slowly relative to the time for the decay of the memory associated with antennation with other ants, the dynamics of both these processes is similar. Then the signals laid down (or transported) by ants increases locally at a rate proportional to their density (*Gordon, 2021*), and is subject to degradation and diffusion slowly. Accounting for these effects, we arrive at the following dynamical equations for the evolution of $\mathbf{r}_j(t)$, $\theta_j(t)$, $c(\mathbf{x}, t)$ as:

$$\dot{\mathbf{r}}_j(t) = \underbrace{v_o \hat{\mathbf{p}}(t)}_{\text{Self-propulsion}} , \tag{1}$$

$$\dot{\theta}_j = \underbrace{G \nabla_{\perp} c}_{\text{Antennation feedback}} + \underbrace{\eta_j(t)}_{\text{Exploration}} , \tag{2}$$

$$\partial_t c = \underbrace{D_c \nabla^2 c}_{\text{Diffusion}} + \underbrace{k + \sum_{j=1}^{n} \mathcal{H}(\mathbf{r}_j(t); a)}_{\text{Production}} - \underbrace{k_- c}_{\text{Decay}} . \tag{3}$$

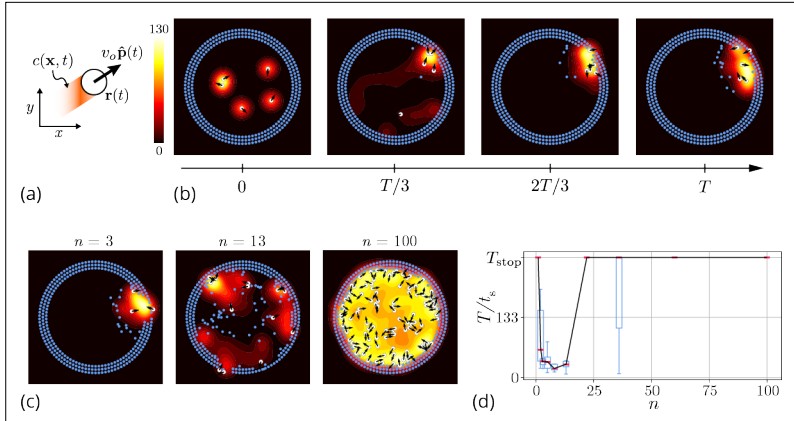

**Figure 3.** Agent-based simulation. (**a**) Schematic of the agents in our simulation captured by their position $\mathbf{r}(t)$ and orientation $\hat{\mathbf{p}}(t)$ moving at speed $v_o$. These agents generate an antennating field $c(\mathbf{x}, t)$ at a constant rate $k_+$ which decays at a rate $k_-$. (**b**) Progression of cooperative excavation of the corral by 5 agents as they pick elements from the boundary and drop them in the interior (see sec. *Appendix 2—table 1* for parameters). Color bar shows the magnitude of antennating field and it varies between 0–130. (**c**) Snapshot of the dynamics at the end of simulations corresponding to $T_{\text{stop}} = 266$ for the number of agents $n = 3, 13, 100$. We see that agents can go from excavating successfully to being trapped in their own communication field. (**d**) Box plot showing the time taken to excavate out of the corral $T/t_s$ (non-dimensionalized using $t_s$ - time taken for an agent to travel the entire domain) as a function of the number of agents $n$ in the corral when $T_{\text{stop}} = 266$. For very small and very large number of agents the collective does not excavate out as the median $T/t_s = T_{\text{stop}}$ and they escape fastest for $n = 8$.

Here, the orientation of the agent in *Equation 1* is given by $\hat{\mathbf{p}}_j = (\cos\theta_j, \sin\theta_j)$ with $\theta_j$ being the heading angle, $v_o$ the characteristic speed of the agent, $\eta_j$ is a Gaussian white noise with correlation function $\langle\eta_j^k(t)\eta_j^l(t')\rangle = 2D_a\delta_{k,l}\delta(t - t')\rangle$. The agents produce an antennating field at a rate $k_+$ which decays at a rate $k_-$ centered around the agent, and captured by the function $\mathcal{H}(\mathbf{r}_j, a) = \{1 \text{ if } |\mathbf{x} - \mathbf{r}_j|^2 - a^2 \leq 0, \text{ and }$ vanishes otherwise$\}$. We assume that the gradient in the antennating field along the local normal, on the right hand side of *Equation 2*, determines the rotation of the agents with $G$ being the rotational gain. In order for the agents to initiate the excavation process, they can pick the elements from the boundary and drop them in the interior of the corral only when the local concentration of the antennating field is larger than a critical threshold $c^*$, consistent with observations (*Gordon, 2021*; *Gordon et al., 1993*). *Figure 3(b)* shows snapshots (see *Video 2* for a movie of the simulations) of the agent-based simulations following *Equations 1–3* showing that the agents excavate successfully out of the corral when the gradient following behavior is strong (see Appendix 2 for details). Given this, we expect the time taken to

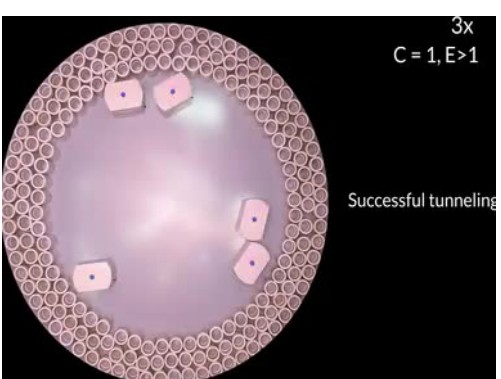

**Video 3.** Successful tunneling in RAnts. (*i*) Dynamics of excavation by RAnts as they cooperatively tunnel through the corral for **C = 1** and without cooperation, **C = 0**; (*ii*) Jammed phase: When the pick-and-place in RAnts is deactivated (corresponding to **E = 0**), they get jammed for **C = 1**; Diffused phase: When the pick-and-place in RAnts is deactivated and the RAnts do not follow the antennating field (corresponding to **C = 0**), they diffuse around.3.

https://elifesciences.org/articles/79638/figures#video3

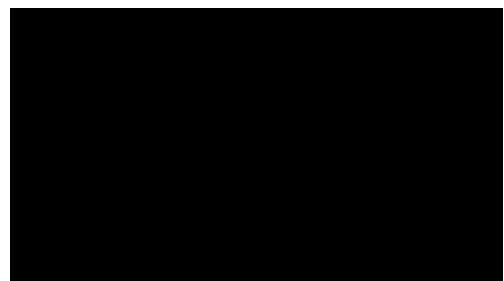

**Video 4.** Summary video showing the results from ant experiments, theoretical model and robot experiments.

https://elifesciences.org/articles/79638/figures#video4

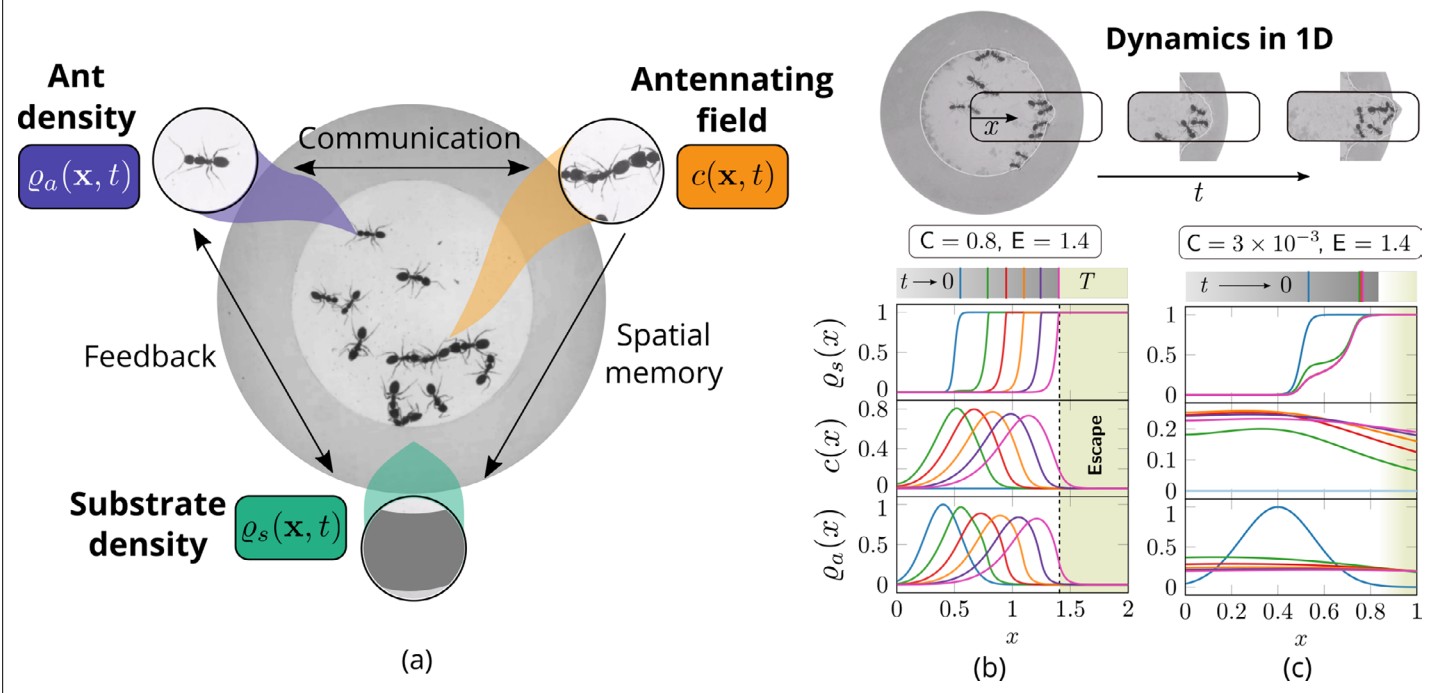

**Figure 4.** Cooperation via organism-environment-organism interaction. (**a**) Schematic of the model showing the interaction between the different spatio-temporal fields required to capture cooperative excavation of ants: ant density, $\varrho_a(\mathbf{x}, t)$; concentration of antennating field, $c(\mathbf{x}, t)$ capturing inter-ant communication; density of corral, $\varrho_s(\mathbf{x}, t)$ representing the soft corral which the ants excavate. We capture the dynamics of excavation by ants close to the excavation site using the one-dimensional version of **Equations 3–5**. (**b, c**) Temporal progression of the corral density, antennating field and the ant density showing successful excavation for high cooperation captured using the non-dimensional number, C (representing non-dimensional strength of cooperation amongst ants) and faster excavation, captured using E. For reduced cooperation ants' diffusion dominates and only partial tunnels are formed (see Appendix 2 for details). $T$ here is the time for excavating out of the corral. The agent density is a gaussian function centered around $x = 0.5$.

escape from the corral is a function of the number of agents. In **Figure 3(c and d)** we see that as we vary the number of agents from $n = 1 - 100$ , for very small or large number of agents in the corral, the agents are unable to escape over the time of simulations, $T_{\mathrm{stop}}$ (ref. **Figure 3(c and d)**), seen as saturation in the excavation time $T/t_s$.

In our agent-based simulations, we can encode the detailed behavior of individual ants and thus account for nuances and variations across the population. However, these simulations are computationally expensive as one needs to couple the dynamics of the antennating field (governed by a partial differential equation) with the motion of discrete agents while also evaluating the mutual interactions between all the agents in the corral. A complementary perspective that allows us to gain insights into the relevant parameters that govern the macroscopic dynamics of the collective is afforded by a theoretical framework that averages over the fast times and short length scale actions of the agents, considering spatial variations over scales much larger than a 'mean-free path' and 'collision time' associated with agent-agent interactions. Our effective theory attempts to couple three slowly-varying spatio-temporal fields: the ant density $\varrho_a(\mathbf{x}, t)$, a communication field $c(\mathbf{x}, t)$ representing antennation and pheromone-based communication, and the corral density $\varrho_s(\mathbf{x}, t)$, shown schematically in **Figure 4(a)**. In the continuum picture, the agents' random motion is captured using diffusion of the density while the rectified motion due to pheromone gradients is captured through chemotaxis, in addition to being self-propelled with a velocity $\mathbf{u}_a$ that is related to the local environment. Finally, motivated by observations of antennation (**Gordon, 1999**; **Pagliara et al., 2018**), we assume that when the ants are stimulated by the presence of the corral past a threshold of antennation, $c^*$ they start excavating. The rate of excavation is assumed to be proportional to the difference in the pheromone concentration relative to the threshold value (see further details). Accounting for these effects,

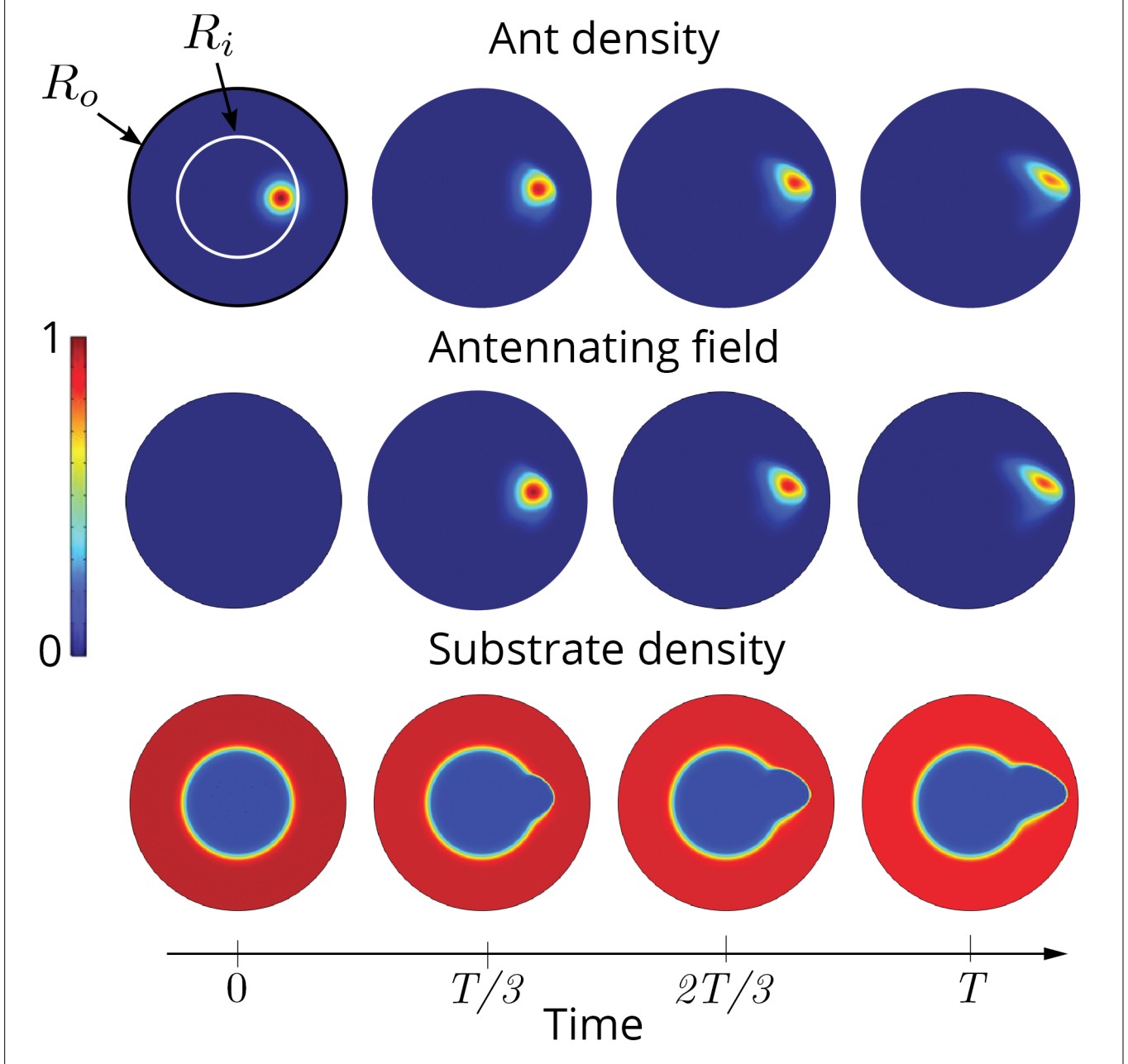

**Figure 5.** Two dimensional simulations showing the evolution of the ant density $\varrho_a$, antennating field $c$ and the corral density $\varrho_s$ by evolving *Equations 3–5*, capturing successful tunneling for non-dimensional numbers $\mathsf{C} = 0.8$, $\mathsf{E} = 1.44$ and time of simulation $T = 20.0$. The list of dimensional parameters used in the simulation are indicated in the *Appendix 1—figure 1(f)*. Radius of the outer boundary, $R_o$ is 5 non-dimensional units and the inner boundary is $R_i = 2.5$ (see Appendix 2 for details). Color bar shows the magnitude of different variables and they vary between 0 and 1.

we arrive at the following dynamical equations for the evolution of $\varrho_a(\mathbf{x}, t)$ and $\varrho_s(\mathbf{x}, t)$ that are coupled to *Equation 3* for the evolution of the communication field:

$$\partial_t \varrho_a = - \underbrace{\nabla \cdot (\boldsymbol{u}_a \varrho_a)}_{\substack{\text{Self-propulsive} \\ \text{advection}}} + \nabla \cdot (\ \underbrace{D_a \nabla \varrho_a}_{\text{Diffusive flux}} - \underbrace{\chi \varrho_a \nabla c}_{\text{Tactile feedback}}\ ), \tag{4}$$

$$\partial_t \varrho_s = -k_s \varrho_s \{\ \Theta \underbrace{(c - c^*)}_{\substack{\text{Antennating} \\ \text{field threshold}}}\ \} \times \{\ \Theta \underbrace{(\varrho_a - \varrho_a^*)}_{\substack{\text{Ant density} \\ \text{threshold}}}\ \}. \tag{5}$$

In *Equation 4*, the ant advection velocity is assumed to have the form $\mathbf{u}_a = v_o(1 - \varrho_s/\varrho_o)\hat{\mathbf{p}}$ where $v_o$ is the characteristic speed of the agents, and $\hat{\mathbf{p}}$ is a unit vector pointing along the radial ($\theta$) direction, and the term $(1 - \varrho_s/\varrho_o)$ reflects the fact that excavating ants are slowed down by their labor; $D_a$ is the diffusivity of ants, $\chi$ is a chemotactic gain associated with the antennating-field-following behavior (related to the gain $G$ in the agent-based model). Here is the average density of the ants defined by where is the domain size. This is a natural scale of the ant density as *Equation 4* is in conservative form and the net density of the ants is preserved over the evolution. In *Equation 5*, $k_s$ is the rate of excavation of the corral and $\varrho_a^*, c^*$ are respectively the threshold concentration of ant density and antennating field required to initiate excavation. We assume that the behavioral switches have simple switch-like responses modeled here via the Heaviside function $\Theta(x)$ (or its regularization via hyperbolic or Hill functions). It is useful to note that in the absence of excavation dynamics, our framework reduces to the well known Keller-Segel model for chemotaxis (see *Hillen and Painter, 2009* for a recent review) (also detailed in Appendix 2). The coupling of ant behavior to the dynamics of excavation introduces the all-important notion of *functional* collective behavior linking active agents, communication channels (the antennating and pheromone fields) and a dynamic, erodible corral that characterizes progress towards task completion.

## Model parametrization and description

The evolution of the ant density in *Equation 4* is a combination of three dynamical processes: ant migration, diffusion and biased motion due to antennating. There are three time-scales associated with these three processes: a diffusion time-scale $\tau_a \sim l^2/D_a$, a collective migration time-scale $\tau_v \sim l/v_o$ and a time-scale associated with taxis $\tau_x \sim l^2/\chi c_o$, where $l$ is a characteristic length-scale. This last scale can be either the width of the corral to be excavated $L$ (which is assumed to be of same order as width of initial ant density profile $l_a$), the length-scale associated with the balance between antennating field diffusion and decay, $l \sim (D_c/k_-)^{1/2}$ or the length-scale due to the advection of ant density and diffusion, $l \sim D_a/v_o$. The antennating field in *Equation 3* is governed by three processes, the generation of the antennating field, as well as its decay and diffusion. This leads to three more time-scales : an antennating field production time-scale $\tau_+ \sim c_o/(k_+\varrho_o)$, a diffusion time-scale $\tau_c \sim l^2/D_c$, and a decay time-scale $\tau_- \sim 1/k_-$. Lastly, the dynamics of excavation from the corral which follows *Equation 5* is governed by a characteristic time-scale $\tau_s \sim 1/k_s$. The list of all seven time-scales and length-scales associated with the different processes in the model are in *Appendix 2—table 2*. In terms of the different time-scales (see Appendix 2 for a list along with their ranges), there are a total of six dimensionless parameters, of which two non-dimensional numbers are qualitatively important in capturing the etho-space of collective excavation: (i) the scaled cooperation parameter defined as $C = \tau_a/\tau_x = \chi c_o/D_a$ which determines the relative strength of antennation (gradient-following) to ant diffusion with $c_o$ being the maximum amplitude of the antennating field, (ii) the scaled excavation rate, $E = \tau_v/\tau_s = k_s l/v_o$. Here, $l/v_o$ is the characteristic time-scale of ant motion, with $l \sim \min[(D_c/k_-)^{1/2}, l_a]$, where $l_a$ is the ant size (see Appendix 2 for details). The other four dimensionless parameters follow from the ratio of the time scale of ant motion and the diffusive time-scale as $V = \tau_x/\tau_a = v_o l/D_a$. The ratio of the rate of production of pheromone and the rate of diffusion or decay, leading to the parameters $\hat{k}_\pm = \tau_-/\tau_+ = k_+\varrho_o/(k_-c_o)$ and $D_c = \tau_-/\tau_c = D_c/(l^2 k_-)$ so that the complete set of non-dimensional numbers that capture the dynamics of the ant collective is given by

$$C = \frac{\chi c_o}{D_a}, \quad E = \frac{k_s l}{v_o}, \quad V = \frac{v_o l}{D_a}, \quad \hat{k}_\pm = \frac{k_+\varrho_o}{k_-c_o}, \quad D_c = \frac{D_c}{l^2 k_-}.$$

In terms of these parameters, the dynamics of the ant density, the antennating field and the corral density given by *Equations 3–5* can be written in non-dimensional form as

$$\partial_t \varrho_a + \nabla \cdot [(C\nabla c + V(1 - \varrho_s))\varrho_a] = \nabla^2 \varrho_a, \tag{6}$$

$$\partial_t c = D_c \nabla^2 c + \hat{k}_\pm \varrho_a - c, \tag{7}$$

$$\partial_t \varrho_s = -\tfrac{1}{4} E \varrho_s (1 + \tanh[\alpha_c(c - c^*)]) \times (1 + \tanh[\alpha_c(\varrho_a - \varrho_a^*)]). \tag{8}$$

> ## Box 1. Ant behavior → Model → Robot behavior
>
> Ants inside the corral move around, communicating with each other using their antennae before they cooperatively excavate the agarose corral. Though the detailed spatio-temporal dynamics of each ant's behavior is different at the microscopic level, we see that the cooperation between the ants results in a persistent density front (see *Figure 1(d, e)* and *Figure 2*) that excavates the substrate. In the theoretical description of the collective's dynamics, the relevant behaviors are encoded through mutual interaction between the ants (via the antennating field) and the substrate. Such a description also inspires the robotic mimics that capture the ant collective's average behavior. We list below the comparison between relevant behaviors in ants and their analogous encoding in the theoretical model as well as in the robots.

To complete the formulation of our model, we also need to specify some initial conditions and boundary conditions for the ant density, the pheromone density, and the location of the corral boundary which are detailed in the Appendix 2.

## Results

### Linear analysis

Before we consider the different limits of the phase-space defined by the non-dimensional numbers, we show that the excavation process is an instability triggered by the scaled excavation parameter $\mathsf{E}$ in the system. Starting with the homogeneous state $\varrho_a^{\mathrm{ss}} = \varrho_a^*, c^{\mathrm{ss}} = c^* = k_+\varrho_o/k_-, \varrho_s^{\mathrm{ss}} = 1$ which satisfies the *Equations 6–8*, and perturbing about this configuration using a plane wave ansatz (in 1D) we write: $\{\varrho_a(x,t) - \varrho_a^{\mathrm{ss}}, c(x,t) - c^{\mathrm{ss}}, \varrho_s^{\mathrm{ss}} - \varrho_s(x,t)\} = \{\tilde{\varrho}_a(k), \tilde{c}(k), \tilde{\varrho}_s(k)\} \exp(ikx + \Omega t)$ where we assume that $\|\tilde{\varrho}_a\|, \|\tilde{c}(k)\|, \|\tilde{\varrho}_s(k)\| \ll 1$. Then the linearized counterparts of the *Equations 6–8* for the ant density, antennating field and the corral density read as: $(\Omega k^2)\tilde{\varrho}_a + ik\mathsf{V}\tilde{\varrho}_s\varrho_o = k^2\mathsf{C}\tilde{c}$ , : $\tilde{c} = \hat{k}_+\tilde{\varrho}_a/(\Omega + 1 + \mathsf{D}_ck^2)$, $\Omega\tilde{\varrho}_s = -\mathsf{E}\tilde{\varrho}_s/2$. From this, we see that the growth rate $\Omega = -\mathsf{E}/2$, is independent of all other parameters in the system, i.e. excavating begins when $\mathsf{E} > 0$, once the ants have created a sufficiently large spatially diffuse antennating field. To understand the dynamics of excavation of the corral and the different phases of collective behavior, we now explore the role of the other non-dimensional numbers.

### Limits of phase-space

Next we discuss the different limits of the phase-space defined by the non-dimensional numbers $\{\mathsf{C}, \mathsf{E}, \mathsf{V}, \hat{k}_\pm, \mathsf{D}_c\}$ and the thresholds $\varrho_a^*, c^*$.

**Table 1.** List of relevant variables and basic behaviors, for ant experiments, theoretical models and robotic implementation.

| Ants | Theoretical model | Robots |
| --- | --- | --- |
| Discrete ants | Ant density, $\varrho_a(\mathbf{x}, t)$ | Discrete robots |
| Antennae communication | Communication field, $c(\mathbf{x}, t)$ | Photormone field |
| Agarose corral | Substrate density, $\varrho_s(\mathbf{x}, t)$ | Boundary elements |
| Motility | Self-propulsive advection, $\mathbf{u}_a$ | Mobile agents |
| Exploratory behavior | Density diffusion, $D_a\nabla\varrho_a$ | Random walk |
| Tactile feedback | Antennating field taxis, $\chi\varrho_a\nabla c$ | Phototaxis |
| Biting behavior | Excavation rate, $k_s$ | Collection and deposition |
| Neural control | Dynamics of ant density | Behavioral rules |

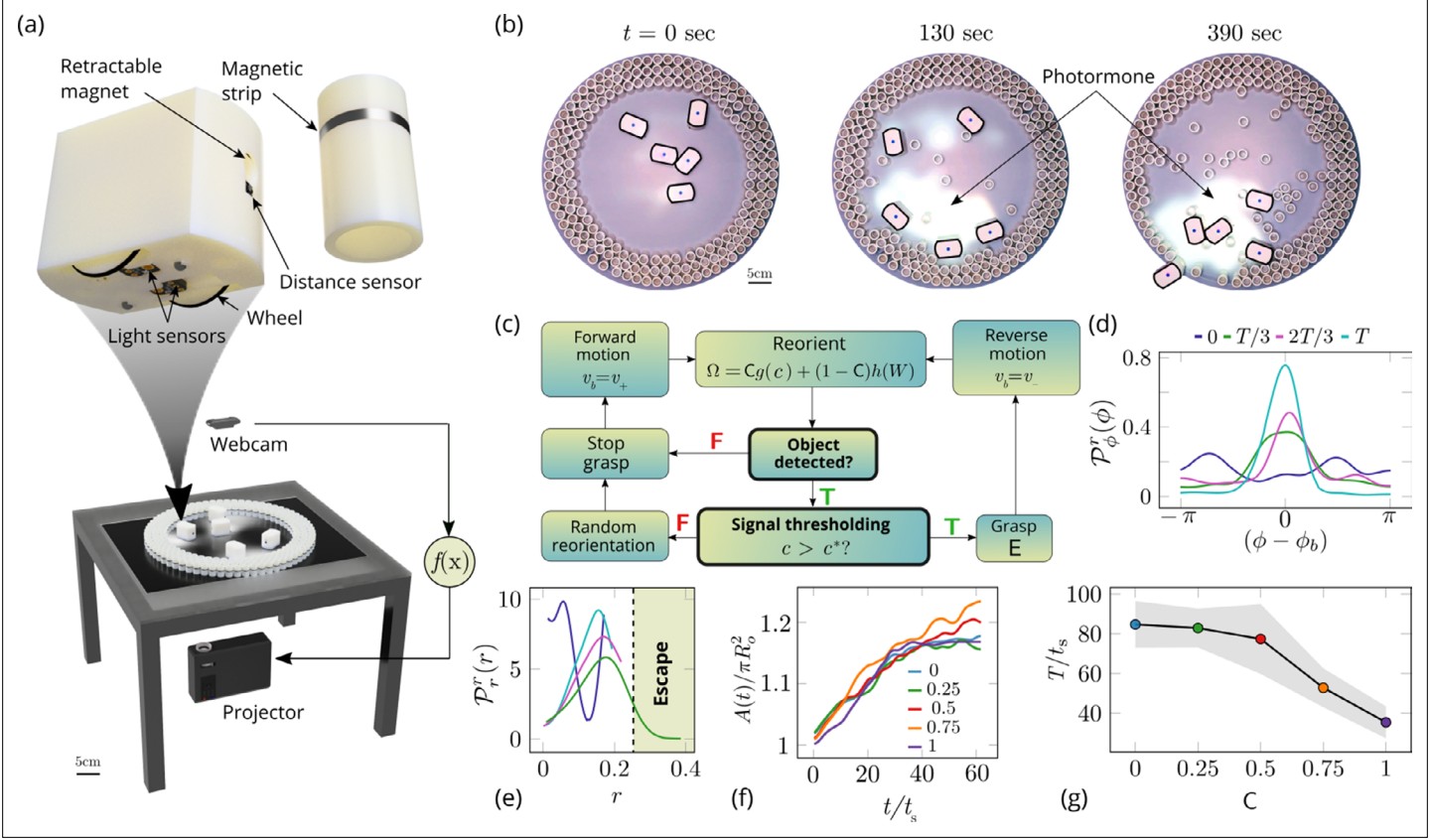

**Figure 6.** Emergent cooperative excavation dynamics in robotic ants. (**a**) Robot Ant (*RAnt*) set-up. A mobile RAnt is placed in an arena 50 cm in diameter surrounded by three layers of cylindrical boundary elements totalling 200 elements. The outermost layer is prevented from being pushed out of the arena by a circular ring. A scalar concentration field (*photormone* field) is projected onto a plane whose intensity can be measured by a RAnt. The position of each RAnt is tracked using a webcam. Each RAnt can pick up and drop the discrete boundary elements using a retractable magnet. (**b**) Series of snapshots at different times of the excavation process for a cooperation parameter C=1. (**c**) Flowchart of the RAnt programming. A base locomotion speed $v_b$ is stored internally and the rate of change $\Omega$ of the heading is a function of the cooperation parameter C, the photormone concentration $c$, and a stochastic process $W$ (Brownian motion). A photormone threshold $c^*$ determines whether an object is grasped (with probability E) after it is detected by the distance sensor. (**d**) Orientation distribution of the RAnt density $\mathcal{P}_\phi^r(\phi, t)$ as a function of the azimuthal position $\phi$ is the orientation of the excavated tunnel. The density is plotted for different times. (**e**) Radial distribution of the RAnt density $\mathcal{P}_r^r(r, t)$ within a sector of $\pi/2$ centered around the position of the excavated tunnel as a function of distance from the center of the arena $r$. The density is plotted for the same times as in (**d**). (**f**) Confinement area $A(t)$ as a function of time, normalized by initial circular confinement with radius $R_o$ for different cooperation parameter C. (**g**) Normalized excavation time $T$ as a function of cooperation parameter C, averaged over 5 experiments per cooperation parameter. Every experiment was run until the first RAnt excavated out or the experiment duration exceeded 15 min.

### Small thresholds, when $\varrho^* \ll \varrho_o$ and $c^* \ll c_o$

When $\varrho_a^* \ll \varrho_o$ and $c^* \ll c_o$, we see the appearance of partial tunneling even with an initially inhomogeneous ant density $\varrho_a$, independent of the pheromone dynamics. However, depending on on the value of the ratio $\tau_s/\tau_v$, the ants can either excavate through the corral completely ($\tau_v/\tau_s \ll 1$) or partially ($\tau_v/\tau_s \lesssim 1$) (ref *Appendix 2—table 3*). If the ants are moving randomly, i.e. in the diffusion-dominated regime, they can still tunnel through the corral if $\tau_c \sim \tau_s$ and partial tunnel through the corral if $\tau_c \lesssim \tau_s$. In non-dimensional terms, this translates to the relations $V \sim \mathcal{O}(1), C \ll 1$ or $V, C \ll 1$ and $E \sim \mathcal{O}(1)$ for the corral evolution. (*Appendix 2—figure 1* shows the results of simulations of both the tunneling and the partial tunneling behavioral phases).

### Cooperation dominated regimes when $C \gg 1$ and $E, V \to 0$

For efficient excavation, the ants need to work collectively by being localized and excavating fast. Spatial localization leads to cooperation via feedback from the antennating field (see *Figure 4(b)*) while for successful excavation, ants need to migrate towards the corral and tunnel through it, so

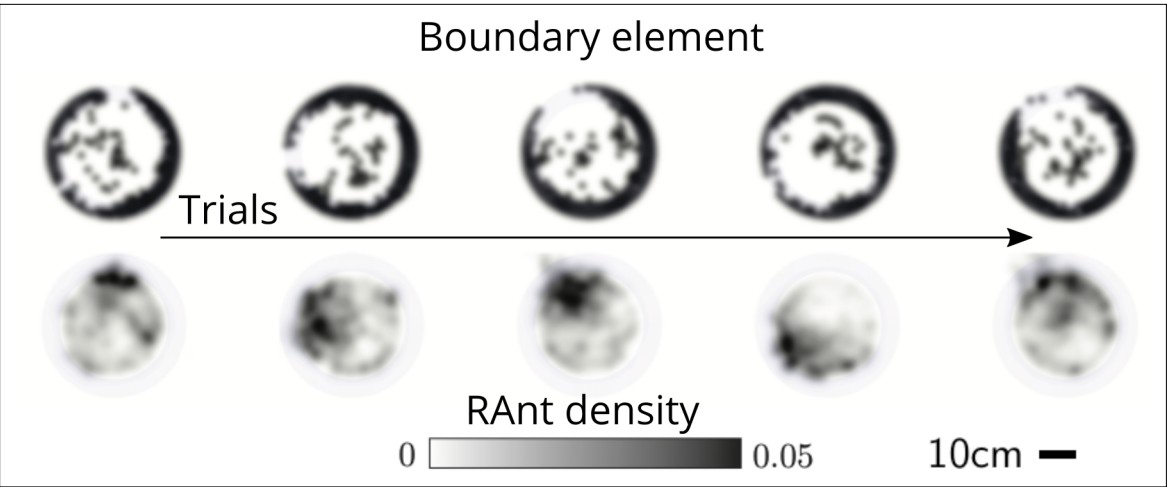

**Figure 7.** Averaged RAnt dynamics.
Ultimate distribution of boundary elements and averaged RAnt density field (in units of #/cm²) over the full duration of experiments for different trials.

that their effective speed $v_o$ needs to be non-zero. To quantify these behaviors, we first look at the dynamics of the ant density and the antennating field in the absence of migration i.e. $V \to 0$ or corral evolution. This leads to three regimes:

- Diffusion dominated regime: When the antennating field diffuses rapidly, i.e. $D_c \sim \hat{k}_\pm \gg 1$, then the equations for the evolution of the antennating field and the ant density, *Equations 3-4* simplify to

$$-D_c\nabla^2 c = k\varrho_a, \tag{9}$$

$$\partial_t\varrho_a + \chi\nabla \cdot (\varrho_a\nabla c) = \boldsymbol{D}_a\nabla^2_{\varrho_a}. \tag{10}$$

- Decay dominated regime: When the antennating field decays fast i.e. $\hat{k}_\pm \sim \mathcal{O}(1), D_c \ll 1$ the dynamics of the antennating field *Equation 3* simplifies to $c \approx (k_+/k_-)\varrho_a$ and the ant density evolution *Equation 4* simplifies to ,

$$\partial_t\varrho_a + \frac{\chi k}{k_-}\nabla \cdot (\varrho_a\nabla\varrho_a) = D_a\nabla^2\varrho_a. \tag{11}$$

- Chemotaxis dominated regime: When the chemocactic coefficient $\chi$ is large, i.e. in dimensionless terms $C \gg 1$, the ant collective gets jammed. To see this we linearize the *Equation 11* about a uniform ant density $\varrho_o$ and recognize that this leads to an effective negative diffusivity and thus a spatio-temporal focusing of the ant density; we leave a detailed analysis of the characteristics of this for future study.

To understand the balance between diffusion of the antennating field and its decay, we note the appearance of a natural length scale $l \sim (D_c/k_-)^{1/2}$ which defines the zone of influence of the field and provides a measure of the non-dimensional tunneling rate indicated in Figure 8. All together, our analysis shows that the dynamics of the antennating field controls the aggregation or diffusion of ant density. But for efficient excavation, especially when the activation thresholds for excavation and localization $\varrho_a^*, c^*$ are large, we need both cooperation and finite velocity of migration. A catalog of the various regimes associated with partial tunneling, jamming, or diffusion as the dimensionless problem parameters are varied is listed in *Appendix 2—table 3*.

To understand how these different limits translate to the dynamics of excavation from the corral induced by the ants, we now consider the case when E, $V \neq 0$, and solve the governing *Equations 4; 5* in a one-dimensional setting (ref Appendix 2). We see that we can capture the two limits of the excavation behavior seen in experiments; for large excavation rates E > 1 and cooperation parameter, C > 1, we see coordinated excavation (shown in *Figure 4(b)* and *Figure 5* in a two-dimensional setting), while decreasing the cooperation parameter leads to disorganized excavation (shown in *Figure 4(c)*) (see *Appendix 2—figure 1*). While a direct comparison with the behavior of ants is not easy owing to the difficulty of inferring the dynamics of information transfer through antennation,

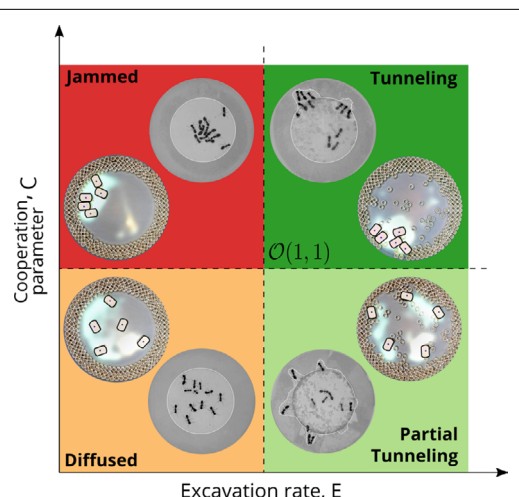

**Figure 8.** Phases of cooperation Phase-diagram of cooperative task execution with different phases seen in ants and RAnts. In the robotic experiments we tune the Cooperation parameter C and the Excavation rate E while in the ant experiments we change the caste mixture. In the ant experiments we see the jammed and diffused phases transiently before the ants relax to cooperative excavation.

the minimal assumptions we have made about the antennating field dynamics suffice to capture the macroscopic behavior of the collective. All together, our agent-based model and the phase-field model shows the emergence of cooperativity without the need for a plan, optimization principle, or internal representations of the world; instead environmentally mediated communication between agents (*Mataric, 1993*) coupled to local behavioral rules suffice to realize robust excavation.

## Robotic collective excavation

Although our quantitative observations of the collective behavior of the ants is qualitatively captured by both our agent-based and continuum models, a natural question we can ask is whether the coarse-grained averaging over of the communication field affects the emergence of the task in experiments, especially since we are unable to measure or directly control the microscopic behaviors of the ants. To go beyond our ability to explain the observations of ant behavior using our theoretical framework, we asked if we might be able to recreate the behavior in artificially engineered mimics, and probe a larger range of the parameters and phase-space spanned by the scaled excavation and cooperation parameters C,E, than our experiments allowed us to - see *Table 1* for a list of the relevant variables across ants, models and robots *Figure 5*.

For this, we turn to a robotic platform to synthesize collective functional behaviors that arise from simple behavioral rules underlying individual programmable robots. Our custom designed robot ants (RAnts) are inspired by many earlier attempts to create artificial agents that are mobile and follow simple rules (*Braitenberg, 1986*; *Brooks, 1991*; *Simon, 1996*), can respond to virtual pheromone fields (*Sugawara et al., 2004*; *Garnier et al., 2007*) and are capable of robotic excavation (*Aguilar et al., 2018*). Our autonomous wheeled robots can exhibit emergent embodied behavior (*Bricard et al., 2013*), and are flexible enough to allow for a range of stigmergic interactions with the environment (*Werfel et al., 2014*; *Petersen et al., 2019*). This is made possible by having each RAnt equipped with an infrared distance sensor to detect obstacles and other RAnts, a retractable magnet that can pick up and drop wall elements with a ferromagnetic ring (shown in *Figure 6(a)*), and the ability to measure a virtual pheromone field generated by a light projected (from below) onto the surface of a transparent arena they operate in (see *Figure 6(a, b)*, *Theraulaz and Bonabeau, 1995*; *Sugawara et al., 2004*; *Garnier et al., 2007*; *Wang et al., 2021*). The intensity of this 'photormone' field follows the antennating field *Equation 2* and thus follows the dynamics of a field that is linked to the locations of the RAnts and diffuses and decays away from it. The photormone field is realized by a projected luminous field on the arena, which the robots can sense. This allows us to use a local form of *Equations 4; 5* to define a robot's behavior in terms of an excavation rate E, a cooperation parameter C, and a threshold concentration for tunneling $c^*$. This is encoded in the behavior-based rules (see *Figure 6(c)* and Appendix 3 for more details), that induces the following behavior: (*i*) follow gradient of projected photormone field; (*ii*) avoid obstacles and other RAnts at higher photormone locations; (*iii*) pick up obstacles from high photormone locations and drop them at low concentration levels. Since the robots have no symbolic representation of the different signals they sense (e.g. they cannot distinguish another RAnt from a wall element, since both merely produce a bump in the sensor signal), the observed behavior emerges from this simple sequence by depending on the current state of the environment and the robot.

Varying the parameter $C \in [0, 1]$ allows us to tune the individual behavior from random motion ($C = 0$) to tracking the photormone gradient ($C = 1$) (see *Video 3*). Varying the non-dimensional excavation rate E changes the frequency at which the robots execute pick-and-drop behavior with detected objects, and serves to mimic what arises in ants as a function of their morphology and caste (see Appendix 1 for more details). For specific values of these parameters, we followed the collective behavior of RAnts by averaging their position over several pick-and-drop timescales to obtain the RAnt density field $\varrho_r(r, \phi, t)$, just as for ants. When all the RAnts are programmed to have a cooperation parameter $C = 1$, RAnts initially explore the region without picking the boundary element until the photormone concentration $c \sim c^*$, which happens once a particular location has enough visits by other RAnts. Just as for ants, we calculate the radially averaged RAnt density $\mathcal{P}_\phi^r(\phi, t) = \int \varrho_r(r, \phi, t) dr$; *Figure 6(d)* shows how RAnt density localizes at a (random) value of the azimuthal angle. As excavation progresses, the RAnt density propagates radially outwards as a density front just as in ants, shown in *Figure 6(e)* in terms of the quantity $\mathcal{P}_r^r(r, t) = \int \varrho_r(r, \phi, t) d\phi$ (also shown in *Figure 7* for different trails when $C = 1$). Concommitantly, as excavation progresses, the corral area increases (*Toffin et al., 2009*); interestingly the scaled corral area $A(t)/\pi R_0^2$ is independent of the cooperation parameter C as shown in *Figure 6(f)* (all RAnts were programmed to have the same excavation rate).

However, cooperation does change the time for excavation; in *Figure 6(g)* we show the average excavation time (scaled by the characteristic time it takes for a rant to traverse the arena) and see that $T/t_s$ decreases with an increase in the cooperation parameter C. RAnts excavated out every time for $C > 0.5$, but are unable to complete excavation for low values of the cooperation parameter (within a 15-min time window). Our results show that it is the localized collective excavation of RAnts mediated by photormone-induced cooperation that is responsible for efficient tunneling and excavation; for low values of C, tunneling is defocused and global, and thus not as effective (see *Appendix 3—figure 2*). When $E \to 0$ (vanishing probability for a successful pick up) but C is large(see *Figure 8* and Appendix 2 for theoretical predictions), the RAnts get jammed because they follow the photormone field they generate but are unable to tunnel through the boundary constriction. On the other hand, when E <1 and C<1 the agents do not cooperate and their diffusive behavior prevents successful tunneling. The range of strategies can be visualized in a two-dimensional phase space spanned by the variables E and C shown in *Figure 8*. Low values of C and E lead to diffusive (and non-functional) behavior, while high values of these variables lead to coordinated excavation, with the other two quadrants corresponding to jammed states (large C, small E) and partially tunneled states (large E, small C). Interestingly, these states are also observed as transients in our ant experiments, for example in the initially diffused state that is characterized by random motion inside the corral, when transiently jammed states and partial tunneling occur (see *Videos 1 and 4*).

## Discussion

Our analysis of collective behavior in a functional task, excavation, uses quantitative observations of ants to build theoretical and computational models to explain them, and recreate these behaviors using a swarm robotic system (see *Video 4* for a summary). Our simple dynamical models involving individual agents as well as an effective continuum theory provide a phase diagram that shows how the transition from an individually exploratory strategy to an exploitative cooperative solution is mediated by the local chemical and mechanical environment. Our study suggested algorithms that we then deployed in an engineered system of robots that individually follow a minimal set of behavioral rules that mould the environment and are modulated by it.; the malleable environment serves both as a spatial memory as well as a computational platform (using the spatio-temporal photormone field and the corral). Our simulations of agent-based models and robotic experiments further suggest that a coarse-grained framework linking behavior, communication and a modulated environment is relatively robust to failure of and stochasticity in the behavior of individual agents (i.e. variations in initial conditions and number of agents), in the communication channels and in the corral geometry, in contrast to engineering approaches that aim to control all agents and optimize costs.

Different strategies such as collective excavation, jamming, and diffusion then arise as a function of the relative strength of the cooperation (representing the ability to follow gradients and detect threshold values) and excavation parameters (representing the ability to move material), as manifested in a phase diagram, and the emergence of cooperation arises due to the relatively slow decay of an environmental signal (the pheromone/antennating/photormone field), coupled to a threshold excavation rate. Since the ability to solve complex eco-physiological problems such as collective excavation is directly correlated with a selective (functional) advantage in an evolutionary setting, perhaps collective behavior must always be studied in a functional context.

## Acknowledgements

We thank the NSF PHY1606895 (SGP, LM), Swiss National Science foundation (FG, grant P400P2-191115), Ford foundation (JK), NSF EFRI 18–30901 (LM), NSF 1764269 (LM), Kavli Institute for Bionano Science and Technology (SM, VM, LM), the Simons Foundation (LM) and the Henri Seydoux Fund (LM) for partial financial support.

## Additional information

### Funding

| Funder | Grant reference number | Author |
| --- | --- | --- |
| National Science Foundation | PHY1606895 | L Mahadevan |
| Henri Seydoux Foundation | | L Mahadevan |
| National Science Foundation | PHY1606895 | S Ganga Prasath |
| Swiss National Science Foundation | | Fabio Giardina |
| Ford Foundation | | Jordan Kennedy |
| National Science Foundation | PHY1764269 | L Mahadevan |

The funders had no role in study design, data collection and interpretation, or the decision to submit the work for publication.

### Author contributions

S Ganga Prasath, Conceptualization, Data curation, Formal analysis, Methodology, Writing – original draft, Writing – review and editing; Souvik Mandal, Fabio Giardina, Conceptualization, Data curation, Formal analysis, Investigation, Methodology, Writing – original draft, Writing – review and editing; Jordan Kennedy, Methodology; Venkatesh N Murthy, Conceptualization, Methodology, Writing – original draft, Project administration, Writing – review and editing; L Mahadevan, Conceptualization, Funding acquisition, Investigation, Writing – original draft, Project administration, Writing – review and editing

### Author ORCIDs

S Ganga Prasath http://orcid.org/0000-0002-4545-911X
Souvik Mandal http://orcid.org/0000-0002-9552-5613
Venkatesh N Murthy http://orcid.org/0000-0003-2443-4252
L Mahadevan http://orcid.org/0000-0002-5114-0519

### Decision letter and Author response

Decision letter https://doi.org/10.7554/eLife.79638.sa1
Author response https://doi.org/10.7554/eLife.79638.sa2

## Additional files

### Supplementary files
• MDAR checklist

### Data availability
All the data used to generate the figures in the article are available here: https://github.com/sgan-gaprasath/rantlFigData (copy archived at swh:1:rev:ba2c6291882cf2355c0fc5d27384a8ce0dc48cc5). The simulation code used in the article is also available in the same folder.

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

## Appendix 1

### Ant experiments

#### Experimental setup - handling ants

We collected two queen-right mature colonies of *Camponotus pennsylvanicus*, established in logs of fallen trees, from the Middlesex Fells Reserve (42.45 °N, 71.11 °W) in August 2019. Each mature colony consists of three morphologically distinct castes of worker ants: major, media and minor, with an average body length of 7 mm, 5 mm for media, and 4 mm respectively. We placed the collected wooden logs housing those colonies in two separate plastic "home" boxes. We coated the inner wall of each home box with ant-slip Fluon to prevent the ants from escaping the home box. Each home box was connected to a foraging box by a tube through which ants travelled to and fro. We kept the whole set up in the laboratory with a 12 hour light-dark cycle, 30°C temperature and 50–70% relative humidity. Before we moved to the next phase of the experiment, i.e. the data collection, we waited for the ants to resume foraging and excavation of woods (for expanding their galleries) inside their home wooden log; this took 3–5 days after the relocation.

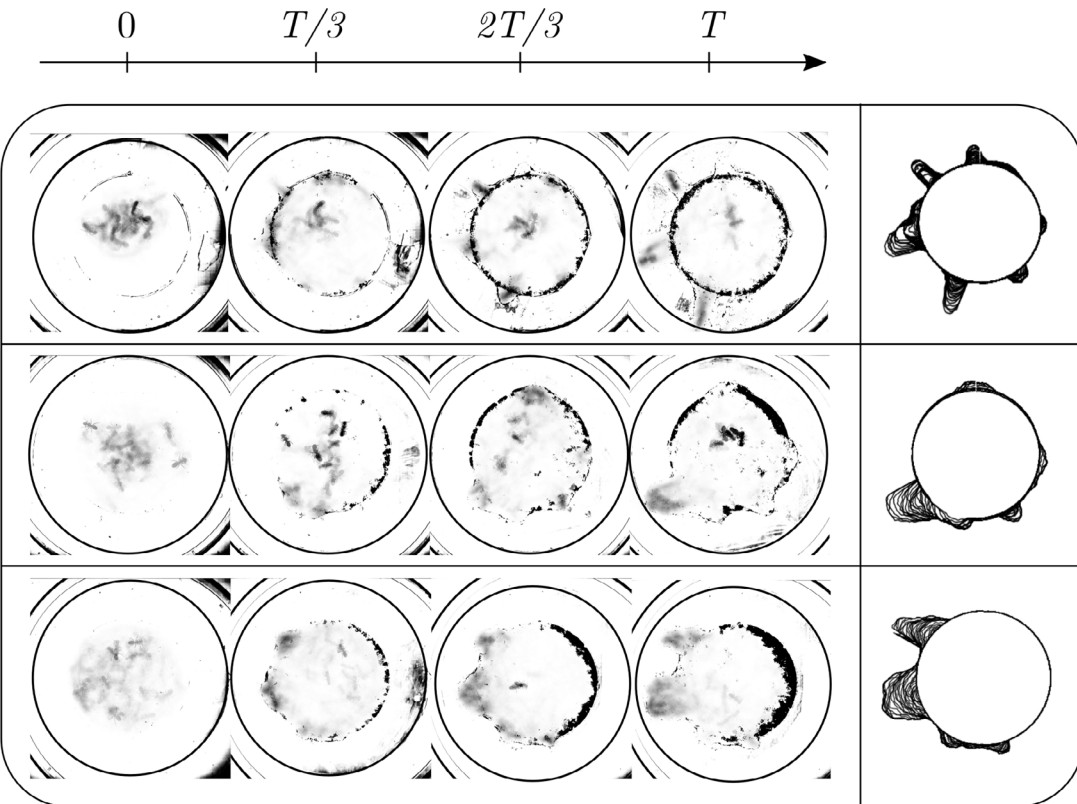

**Appendix 1—figure 1.** Dynamics of ant density field, $\varrho_a(\mathbf{x}, t)$ (in units of #/mm²) obtained by averaging the ant location and the boundary shape $R(\phi)$ when 4 ants each of major, media and minor types are confined inside the agar ring for different trials.

About 10 minutes prior to the experiments, we collected ants engaged in wood excavation from the surface of the nest log. We used insect aspirators for collecting the ants. Once we collected all ants needed for the experiment, we subjected the ants to Carbon dioxide anaesthesia for 1 minute. Next, we placed the anaesthetised ants in the agarose well in the experimental arena; we placed each ant at least 1 cm away from any other ant. Ants regained their activity in the next 5–10 minutes.

#### Experimental setup - confinement

For the next phase of the experiment, we needed to confine the ants in an excavatable enclosure. This is the corral that the ants need to bite through to free themselves. We used a ring-like confinement made of agarose gel, with a height of 10 mm, an inner radius of 35 mm and outer radius of 55 mm, making the ring 20 mm thick. To make a precise shape of the ring repeatedly, we custom-built a

casting mold made of acrylic plastic. We started preparing for the Agar ring before we collected the ants. For making the ring, first, we thoroughly mixed $3\,\mathrm{gm}$ of Agar powder in $100\,\mathrm{ml}$ of tap water. We then warmed the solution in a microwave oven until the solution started bubbling and appeared clear. Next, we poured the solution in the plastic mold, and kept it in 30 C temperature for 25 minutes; the agarose gel solidified and become opaque during this time. Once the agarose turned solid, we placed the ring on top of plastic sheet in the arena. Next, we placed the ants inside the ring and put a perti dish lid on top of the agarose ring. Thus, we confined the ants - with a solid plastic floor and ceiling, and an excavatable agarose gel wall. A schematic of the set-up is shown in *Figure 1(b)*.

## Experimental setup - arena and video recording

The arena consists of a piece of white 3 mm thick plastic sheet as the substratum, illuminated with infrared back-light, and surrounded by a 1.5 cm high plastic wall coated with Fluon ant-slip. We placed a Point Grey (FLIR) Grasshopper3 GS3-U3-41C6NIR camera, fitted with a 65 mm macro lens, on top of the arena to capture the view of the whole ring. The camera recorded the videos with 30 fps recording speed and 1024×1024 pixels resolution.

## Markerless tracking

Leveraging an open source, deep-learning based pose estimator package SLEAP (*Pereira et al., 2020*), we track 3 body parts in each ant - head, thorax, and abdomen (gaster). Sample results obtained from this tracking is shown in *Appendix 1—figure 2(e)* and in $(f - h)$ we quantify the noise statistics of ant motion and its orientation using the tracking data. Ants initially move randomly in the confinement and one of the ants starts the excavation process after which several ants start excavating cooperatively at the same location. When the tunneling happens, all the ants are orientated along the tunnel. We see this through the progression of the orientation distribution of ants $\mathcal{P}(\theta, t)$ in *Appendix 1—figure 2(j)*. To characterize the localization in ant orientation as the excavation proceeds, we use a von-Mises distribution (the analog of a Gaussian distribution for a periodic variable, given by $\mathcal{P}(\theta, t; \mu(t), K(t)) = \exp\left[K\cos(\theta - \mu)\right]/2\pi I_0(K))$ of the ants (where µ is the mean local orientation associated with location of tunnel along the boundary). In *Appendix 1—figure 2(k)*, we see that over time, $K(t)$ increases, i.e. the variance decreases. During the excavation process, ants bite through the corral, carry the debris from the excavation site and drop it in the interior of the confinement. This happens over and over again until all the ants excavate out. We see this captured in the oscillations of the location of ants as shown in *Appendix 1—figure 2(i)*.

## Average dynamics

We have a total of 7 sets of experiments with four sets of experiments with a collective of 12 majors ants and 3 sets of experiments with a mixture of 4 major, 4 media and 4 minor ants. Using the recorded video of the ant excavation dynamics, we threshold the intensity to extract only the ant boundary and average the ant dynamics over 250 secs. This gives us a density field of ants representing the locations where the ants have been and the amount of time they spend. We found in our experiments that each ant bites the corral, picks the bitten piece and transports it into the interior of the confinement. This process takes approximately 60 secs (see *Appendix 1—figure 2(i)*) and we would like to average the ant dynamics over several 'turn over' time-scales. We chose 250 secs and the obtained density field is shown in *Appendix 1—figure 2*. We perform this averaging for the experiments with all major ants as well as the mixture of different castes. In all the experiments, an ant density front propagates through the corral as they excavate and gradually tunnel through.

## Boundary tracking

From the recorded videos, we also track the locations in which the ants excavate for creating the tunnel. For that, we used a custom image processing Matlab script. First, we created a mask superimposing on the area encircled by the inner ring of the corral; we colored the mask with a shade different from the corral. When ants excavated the corral, the Matlab script could detect the difference in the shade/color of the excavated area. Using this contrast, we track the continuously changing boundary.

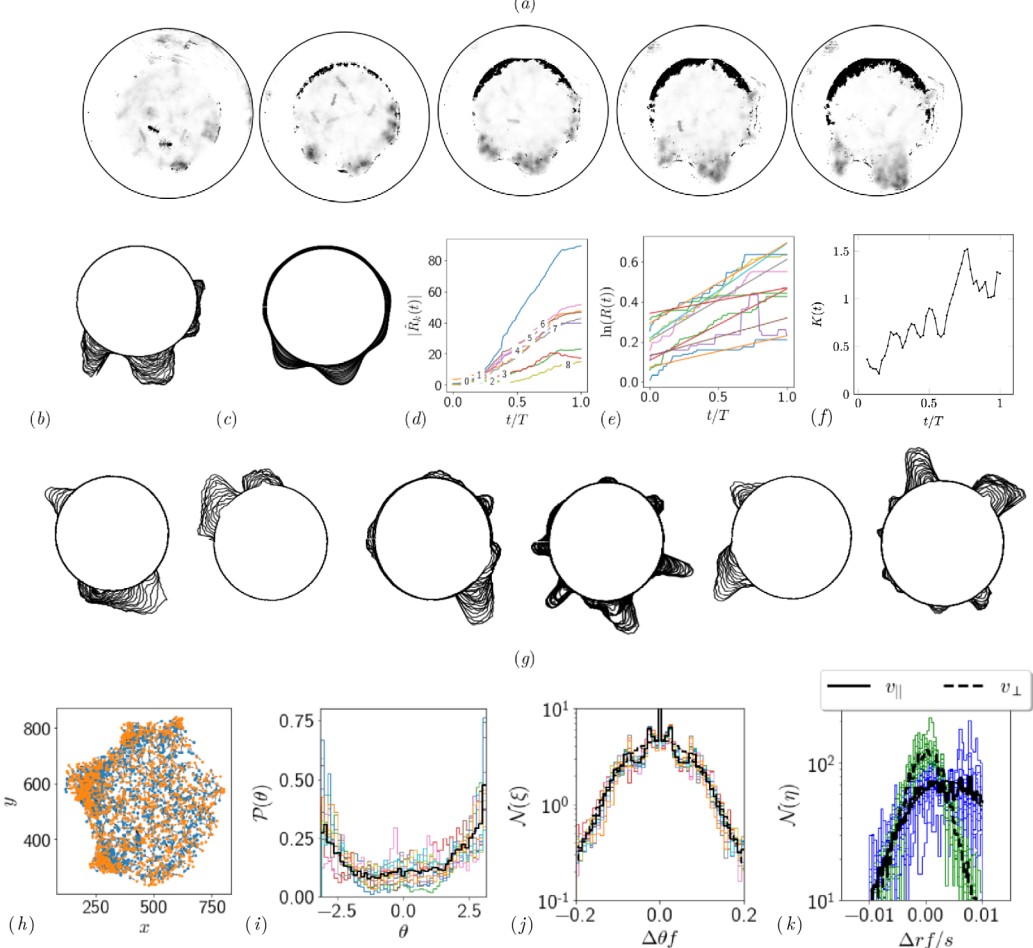

**Appendix 1—figure 2.** Quantifying competition between different modes of exploration and the transition towards exploitation. (**a, b**) Shape of the boundary tunnel during the tunneling process and the approximate representation of the shape using first 9 Fourier modes. (**c**) Evolution of the magnitude of the first 9 Fourier modes of the boundary: $R(\phi, t) = \sum_k \hat{R}(k, t) e^{ik\phi}$. (**d**) Evolution of boundary location, $R(t)$ at different $\phi$ values and the excavation rate. (**g**) Collage of boundary evolution showing tunnel formation in six experiments. (**e**) Location and orientation of the ants obtained from tracking the gaster (orange) and heat (blue) of all the ants. (**f**) Is the von Mises parameter highlighting strength of focus in ant orientation thorough a fit to the $\mathcal{P}(\theta)$ obtained by a curve fit to the distribution. (**g**) Image showing evolution of the boundary as the excavation process happens for different experimental trials. (**h**) Location of center of ants with orientation during the excavation process. (**i**) Average orientation distribution $\mathcal{P}(\theta)$ of all the ants showing hints of localization which is evident when plotted over time. (**j**) Noise statistics of ant velocity along the body axis, $v_{\parallel}$ and in the normal direction, $v_{\perp}$. Dashed lines again are Gaussian fit to the data. Ants have zero mean velocity normal to its axis. (**k**) Noise statistics of orientation with peak close to 0 because of resting of the ants which otherwise follows a Gaussian which is the dashed line.

This is shown as a super-imposed image on the right side of *Appendix 1—figure 2* where $R(\phi, t)$ is the radius of the ring as a function of the polar angle $\phi$. Tunnels are locations along $\phi$ which see increase in the radius. We quantify this by plotting $R(t)$ in *Appendix 1—figure 2(d)*. We also quantify the number of tunnels by decomposing the shape into different Fourier modes as detailed in the caption.

# Appendix 2

## Agent-based model of cooperative task execution

The results shown in *Figure 3* are based on a numerical simulation where discrete agents operate in a continuum scalar communication field, and governed by *Equation (1)*, *Equations 2; 3* . Some additional behavioral rules have to be defined to model the interaction of agents with the substrate. We realize the substrate by discrete obstacles arranged in a circular ring. Agents will attach to an obstacle if they are within the detection range, $l_d$, and if the measured communication field value is above the threshold, i.e. $c \geq c_{hi}^*$. The agent will then reverse its direction of motion, by changing the sign of $G$ in *Equation (2)*. This results in a gradient descent behavior and the attached obstacle will be detached once the measured communication field value satisfies $c < c_{lo}^*$. After detachment, the sign of $G$ is changed again. If agents encounter other agents or obstacles within the detection radius but $c < c^*$, the agents will avoid the obstacle by turning randomly.

There are a few tuned behaviors we implemented to allow scaling the simulation to larger numbers of agents while maintaining the tunneling behavior. First, the gradient $\nabla_\perp c$ in *Equation (2)* is is multiplied by a sigmoid function to limit the turning rate of the agents. Second, the noise term in *Equation (2)* was set to zero for this simulation and the only source of randomness are the random turns during obstacle avoidance. Third, agents pause for $t_{p1}$ when they encounter an obstacle and for $t_{p2}$ when picking up an obstacle. This helps disrupting potential "pheromone traps" to be formed where agents are bound to a region of space due to a high field concentration.

The simulation parameters are described in the following table. All parameters are non-dimensionalized by the corral size $L$ as a natural length scale and the base speed of the agents, $v_0$ as a natural speed.

**Appendix 2—table 1.** Parameters of agent-based simulation.

| Parameter | Description | Value |
|---|---|---|
| $n_r$ | Number of agents | 1–100 |
| $n_o$ | Number of substrate elements | 300 |
| $n_l$ | Number of corral layers | 3 |
| $T$ | Maximum simulation time | 66 |
| $k_+$ | Communication field production rate | 97.5 |
| $k_-$ | Communication field decay rate | 0.75 |
| $D_a$ | Communication field diffusivity | $4.2 \times 10^{-3}$ |
| $c_{hi}^*$ | Excavation threshold | $\frac{1}{2}\frac{k_+}{k_-}$ |
| $c_{lo}^*$ | Detachment threshold | 0.11 |
| $\sigma_g^2$ | Agent field production width (variance) | $2.8 \times 10^{-3}$ |
| $l_d$ | Agent obstacle detection range | 0.03 |
| $t_{p1}$ | Pause after obstacle detection | 0.07 |
| $l_{p2}$ | Pause after substrate attachment | 0.27 |
| $G$ | Rotational gain | 0.135 |

## Continuum model of cooperative task execution

The dimensional equations for the ant-density $\varrho_a(\mathbf{x}, t)$, antennating field $c(\mathbf{x}, t)$ and the corral $\varrho_s(\mathbf{x}, t)$ are given by,

$$\partial_t \varrho_a + \nabla \cdot (\boldsymbol{u}_a \varrho_a) = \nabla \cdot (D_a \nabla \varrho_a - \chi \varrho_a \nabla c), \tag{12}$$

$$\partial_t c = D_c \nabla_c^2 + k_+ \varrho_a - k_- c, \tag{13}$$

$$\partial_t \varrho_s = -\frac{1}{4} k_s \varrho_s (1 + \tanh[\alpha_c(c - c^*)])(1 + \tanh[\alpha_c(\varrho_a - \varrho_a^*)]), \tag{14}$$

where velocity of the collective is $\mathbf{u}_a = v_o(1 - \varrho_s/\varrho_o)\hat{\mathbf{p}}$, capturing the reduction in velocity as the ant collides with the corral. We approximate the Heaviside function, $\Theta(x)$ here using the hyperbolic function $[1 + \tanh(x)]/2$. In the coarse-grained picture describing the collective tunneling seen in experiments the relevant variables (shown schematically in *Figure 4*) are the density of ants, $\varrho_a$; their velocity, $\mathbf{u}_a$; amplitude of the antennating field, $c$; the density of corral, $\varrho_s$. Here we discuss the limits of phase-space that are not described in the main text i.e. when $E, \neq 0$ and also the simulation details.

**Appendix 2—table 2.** Time-scales and length-scales associated with different processes in the model in *Equation 12*- *Equation 14*.

| Time-scale | Process |
| --- | --- |
| $\tau_v \sim l/v_o$ | Ant collective migration |
| $\tau_x \sim l^2/(\chi c_o)$ | Taxis due to antennating field gradient |
| $\tau_+ \sim c_o/(k_+\varrho_o)$ | Antennating field production |
| $\tau_c \sim l^2/D_c$ | Antennation field diffusion |
| $\tau_- \sim 1/k_-$ | Antennating field decay |
| $\tau_s \sim 1/k_s$ | Corral excavation |
| Length-scale | Process |
| $L$ | Corral width |
| $l_a$ | Initial width of ant density |
| $D_a/v_o$ | Ant density advection-diffusion |
| $(D_c/k_-)^{1/2}$ | Antennating field diffusion-decay |
| $\tau_a \sim l^2/D_a$ | Ant diffusion |

## Limits of phase-space when $E \neq 0$

Different phases of task execution/failure arise when the excavation parameter $E$ and the cooperation parameter $C$ are varied. In the cooperation dominated phase if the excavation rate of the agents is small, they get jammed and the analysis in the previous section holds true. When the cooperation among the agents is low, we have $C \ll 1$ which results in diffusion dominated regime. Based on the strength of the excavation parameter $E$, the corral can be partially tunneled or just diffuse. Since we assume that the relevant length scale is of the same order as the width of the corral, $L \sim l$, our analysis reduces to different phases based on whether $E \gg 1$ (where we get partial-tunneling) or $E \ll 1$ (we get diffusion). Based on this we get partial tunneling or diffused phase as listed in Appendix 2—table 3. In *Appendix 2—figure 1* we show results from 1-D simulations highlighting the effect of different terms we have discussed from *Equations 4 and 5* corresponding to different parts of the phase space of cooperative excavation. In the ant density diffusion dominated regime, i.e. $C \ll 1, E \ll 1$, shown in *Appendix 2—figure 1(b)*, there is little cooperation; rapid diffusion with slow excavation results in no tunneling. As we have seen in *Figure 4(b, e)*, tunneling and partial tunneling are inferred through the ultimate state of the corral and the ant-density. In *Appendix 2—figure 1(d, e)* we show how the relative rate of the antennating field diffusion compared to decay, i.e. $D_c \sim \mathcal{O}(1)$ leads to either tunneling or partial tunneling as we vary the cooperation parameter $C$. Decreasing $C$ causes the maximum of $\varrho_a(x, t)$, $c(x, t)$ to go down (ref *Appendix 2—figure 1(c)*),and the width of the initial ant density field increases. Increasing $C$ leads to successful tunneling driven by the propagation of the location of maximum ant density $x_{\text{loc}}$ due to excavation of the corral. Furthermore, we see that the ants can be jammed either because the antennating field diffusion dominates, i.e. $D_c \sim \hat{k}_\pm \gg 1$, or because of the same field decays rapidly, i.e. $\hat{k}_\pm \sim \mathcal{O}(1), D_c \ll 1$. In both these cases however cooperation is what drives the aggregation. Lastly, we see that in order to achieve partial tunneling there are several routes depending upon the relative magnitudes of $\{C, E, V, \hat{k}_\pm, D_c\}$ listed in Appendix 2—table 3.

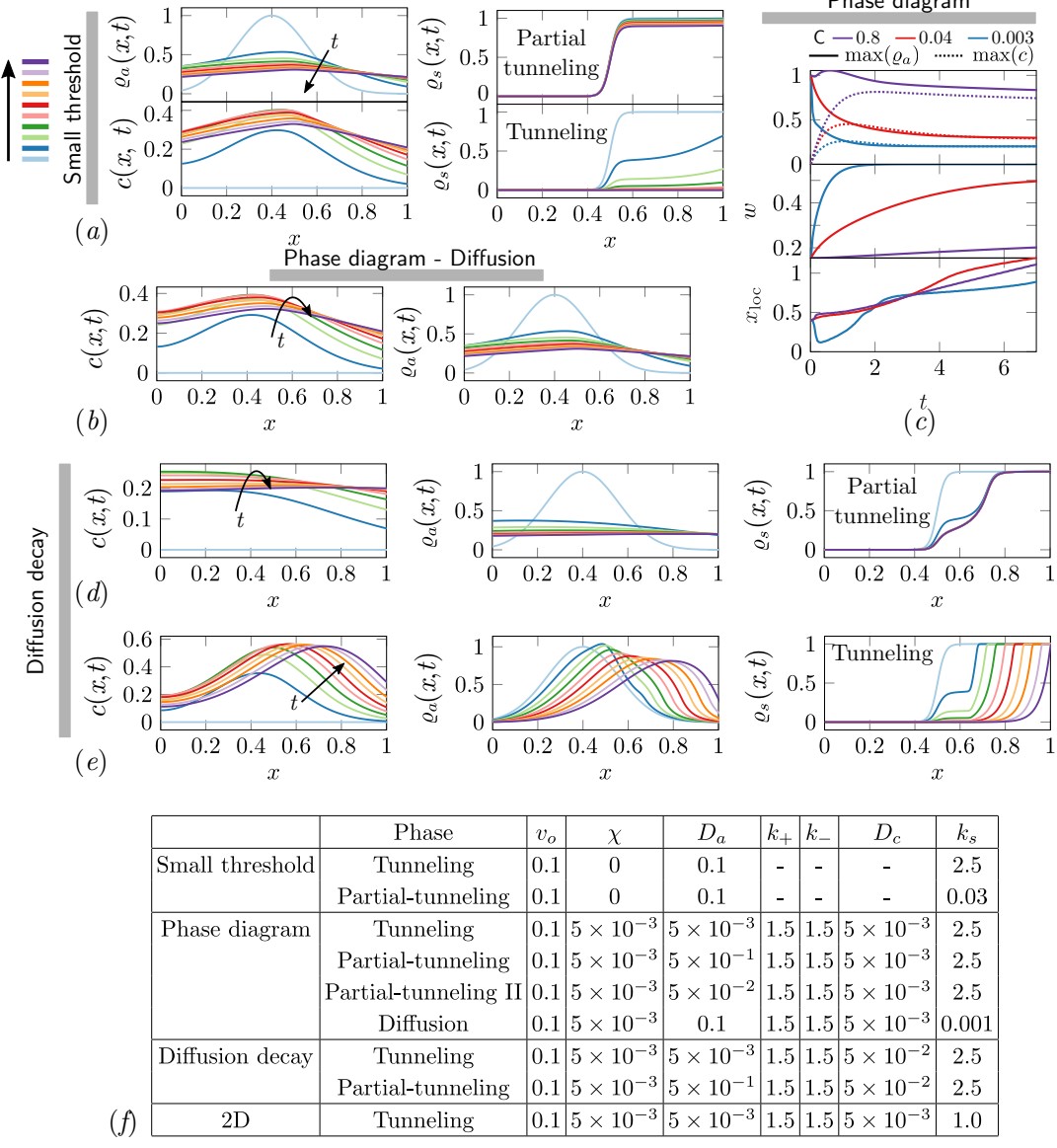

**Appendix 2—figure 1.** The ant density field, $\varrho_a(x,t)$ antennating field $c(x,t)$ and corral density $\varrho_s(x,t)$ for various scenarios of interest in the phase-space. (**a**) Partial tunneling and tunneling when the threshold for excavation is small i.e. $\varrho_a^* = c^* = 0.01$, we see homogeneous excavation and can get tunneling and partial tunneling; (**b**) when we are in the diffusive phase where ant density diffusion dominates, $C = 0.02$ and the excavation rate is very small, $E = 6 \times 10^{-4}$; (**d, e**) partial tunneling and tunneling when the length scale due to antennating field diffusion and decay is of the same order as the initial ant density i.e. $l_a \sim (D_a/k_-)^{1/2}$; (**c**) Evolution of maximum value of $\varrho_a, c$ for 3 different C and fixed $E = 1.44$. (**f**) Table with parameters used in simulations corresponding to different titles shown in gray bar in $(a - e)$

**Appendix 2—table 3.** Different phases in different limits of phase-space of parameters in the model.

| C | E | V | $\hat{k}_\pm$ | $D_c$ | Phase |
|---|---|---|---|---|---|
| $\gg 1$ | $\gg 1$ | $\gg 1$ | $\mathcal{O}(1)$ | $\ll 1$ | Tunneling |
| $\gg 1$ | $\gg 1$ | $\gg 1$ | $\ll 1$ | $\mathcal{O}(1)$ | Tunneling |
| $\ll 1$ | $\gg 1$ | $\gg 1$ | $\mathcal{O}(1)$ | $\ll 1$ | Partial-Tunneling |

*Appendix 2—table 3 Continued on next page*

*Appendix 2—table 3 Continued*

| C | E | V | $\hat{k}_\pm$ | $D_c$ | Phase |
|---|---|---|---|---|---|
| $\ll 1$ | $\gg 1$ | $\gg 1$ | $\ll 1$ | $\mathcal{O}(1)$ | Partial-Tunneling |
| $\gg 1$ | $\ll 1$ | - | $\mathcal{O}(1)$ | $\ll 1$ | Jammed |
| $\gg 1$ | $\ll 1$ | - | $\ll 1$ | $\mathcal{O}(1)$ | Jammed |
| $\ll 1$ | $\ll 1$ | - | - | - | Diffused |

## Simulation details

All the simulations shown in the main text as well the ones above were performed using commercial software COMSOL$^{TM}$, in their general form Partial Differential Equations solver. We choose a very fine resolution with maximum mesh size of 0.005 in a domain of size 2 units in 1D simulations and maximum mesh size of 0.25 in a circular domain of radius 5 units in 2D. The initial condition for the ant density, $\varrho_a(r, 0)$ is $\exp(-(r - r_o)^2/2l_a^2)$ where $r_o = 0.4, l_a = 0.16$ and the density of the corral $\varrho_s(r, 0)$ is chosen to be $[1 + \tanh(\alpha(r - 2.5))]/2$ where $\alpha = 30$. We set the parameter $\alpha_c = 50$ in 1D, $\alpha_c = 10$ in 2D while $\varrho^* = 0.3, c^* = 0.01$. The other parameters used in the simulations in *Figure 4*, *Figure 1* and *Figure 5*. $(a - e)$ are listed in $(f)$. In the 2D simulations in *Figure 5* we assume a spatio-temporally varying self-propulsive velocity field of the form, $\mathbf{u}_a = v_o\{\exp(-y^2/2\sigma^2), \exp(-t/\tau)(1 - \exp(-(x - x_o)^2/2\sigma^2))\}, v_o = 0.1, x_o = 0.2, \sigma^2 = 0.75, \tau = 10$.

## Appendix 3

### Robot Ants

#### RAnt design

RAnts were designed to accommodate the essential electronic and electromechanical parts required for locomotion, picking and placing, and sensing. An exploded view is shown in *Appendix 3—figure 1*. RAnts are powered with a rechargeable 3.7 V battery with 400mAh (Pkcell LIPO 801735) and are coordinated with a microcontroller (Adafruit ItsyBitsy M0 Express). The RAnt's wheels have a diameter of 25 mm and are directly driven with two brushed DC motors with a planetary gearbox rated at 85 RPM at 3.7 V. Rubber o-rings are attached to the wheels to increase traction. A dual motor controller (Pololu DRV8835 Dual Motor Driver Carrier) sets the desired output speed of the motors given a PWM signal from the microcontroller. The mechanism to pick up wall elements was realized using a permanent magnet that is retractable inside the RAnt. A linear servo motor (Spektrum SPMSA2005) moves a permanent magnet inside a guide such that, when fully extended, the magnet attracts ferromagnetic materials and when retracted, the magnetic force is small enough to drop any previously attached objects.

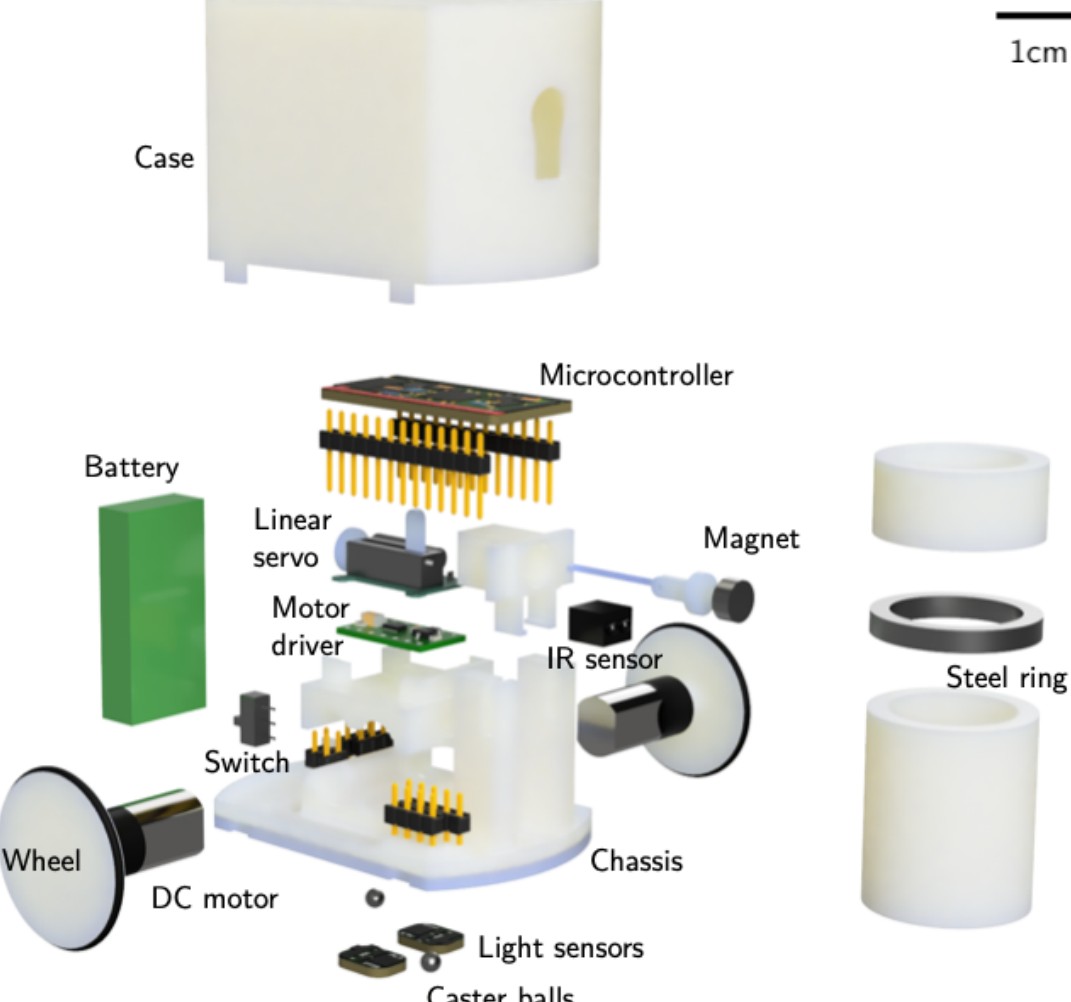

**Appendix 3—figure 1.** Exploded view of a RAnt and a wall element.

The objects to be picked up are cylindrical wall elements of dimension 22mm×40mm made of polyvinyl chloride (PVC) tubes which have a ferromagnetic ring of 3 mm thickness embedded in them. The ring was 3D printed using polylactic acid (PLA) mixed with steel powder (colorFabb SteelFill) and was sandwiched between two PVC tubes. If the RAnt is sufficiently close (≈3 mm) to a wall element with an engaged magnet, the ring in the wall element is attracted to the magnet and

the wall element is slightly lifted from the ground (≈1 mm) for transportation due to the elevated position of the magnet relative to the ferromagnetic ring. RAnts have two typers of sensors; two light sensors (Adafruit ALS-PT19) located at the bottom left and right of the RAnt (relative to the direction of travel) and an infrared (IR) distance sensor (Everlight ITR20001 opto interrupter) capable of detecting objects within 3 cm in front of the RAnt. The chassis of the RAnt is 3D printed using acrylic styrene acrylonitrile (ASA) and supports all the internal components. The wheels of the RAnts were printed with the same material. Due to the design of the wheel arrangement, which was inspired by the zooid robots (*Goc, 2016*), we require two small steel caster balls of 3 mm in diameter that help stabilize the RAnt. The steel balls can be pressed into the bottom of the 3D printed chassis. A 3D printed case made of ASA encloses all internal components of the RAnts, except for a small switch to power the RAnt on or off. A small blue sticker of 6 mm in diameter was placed on the center top of the case and is used for tracking of the RAnt's position with the webcam mounted above the arena.

## RAnt programming

The RAnt behavior is coordinated by the microcontroller which we programmed according to the pseudocode shown in Algorithm 1. The program is initialized with a variable $d$ that encodes the direction of travel (1 for forward, –1 for backward), the cooperation parameter $C \in [0, 1]$, the RAnt's base speed $v_b$, and the light intensity threshold $c^*$. This threshold was set at 50% of the maximal light intensity that can be generated by the photormone field multiplied by the cooperation parameter, i.e. $c^* = 0.5 \times c_{max} \times C$.

After initialization, the program enters a while loop which is running until the RAnt is switched off or the battery voltage drops below 3.5 V. The loop starts with setting the heading of the RAnt, which effectively sets the turning rate. The turning rate is a function of the cooperation parameter and a stochastic process $W$ (Wiener process) which is integrated in the microprocessor. The turning rate follows the equation

$$\Omega = C d \frac{c_L - c_R}{c_{max}} + (1 - C) b \sin(\pi W) \tag{15}$$

with $c_L$ and $c_R$ the photormone intensity measured in the left and right light sensors, respectively, $c_{max}$ is the maximal photormone intensity measurable by the sensors, and $b = 0.3$ is a fixed amplitude. Using a sine function we map the stochastic process $W$ to the range $[-1, 1]$ to avoid getting stuck in constant rotation for large excursions of $W$. The first term in *Equation 15* corresponds to phototaxis using the projected photormone and the second term to a random walk. We can tune the influence of either terms with the cooperation parameter $C$ from pure phototaxis at $C = 1$ to a random walk at $C = 0$. The turning rate is used to define the rotation speed of each wheel. One wheel is always turning at a base rate $\omega_1 = \omega_b = v_b/R$ (with $R$ the wheel's radius) and the other wheel at

$$\omega_2 = \omega_b (1 - 2\|\Omega\|). \tag{16}$$

The assignment of $\omega_1$ and $\omega_2$ to the left and right wheel is flipped according to the sign of $\Omega$. With this definition, at a value of $\Omega = \pm 1$ a RAnt turns on the spot without any translation and at $\Omega = 0$ the RAnt moves on a straight path without rotation.

After the heading was defined and the turning rates sent to the motor driver, the distance sensor is checked for any obstacles that are present up to 3cm in front of the RAnt. At the same time, the light sensors are checked and compared to the threshold value $c^*$. If an object is detected and the photormone value exceeds $c^*$, the RAnt performs a fetching manoeuvre that consists of engaging the magnet with probability E, moving forward for a second with half the base speed $v_b$ then move backwards for the same amount of time. After the fetching manoeuvre, the direction parameter is inverted, i.e. $d = -1$. If an object is picked up with the magnet after the fetching maanoeuvre, the distance sensor will report a detected object as long as it is attached to the magnet. Since $d = -1$, the RAnt will perform the same type of gradient driven locomotion described in *Equation 15* and *Equation 16* but the sign of the signal sent to the motor driver will be inverted, resulting in a reverse motion of the RAnt. If an object is detected, but the photormone concentration in both sensors is lower than $c^*$, an avoidance manoeuvre is performed which consists of a random rotation in place in any direction with the intent to turn away from the detected obstacle.

The next if-statement checks again if an obstacle is detected, but without the condition that the direction parameter is equal to one. If no obstacle is detected, the direction parameter $d$ is set to one and the magnet is disengaged.

---

**Algorithm 1:** RAnt behavioral algorithm

---

**Result:** Cooperative escape in Robot Ants
$d = 1$;
$C \in [0, 1]$;

```
while true do
    set heading;
    if object detected & d = 1 then
        if c > c* then
            engage magent with probability E;
            fetch object;
            d = −1;
        else
            turn away from object;
        end
    end
    if no object detected then
        d = 1;
        disengage magnet;
    end
    if d = 1 and P < kC then
        disengage magnet;
        turn away from object;
        d = 1;
    end
end
```

---

This guarantees that if a fetching manoeuvre is performed but the wall element was not picked up or the object was another RAnt, the RAnt goes back to moving forward.

The last if-statement checks whether the RAnt is in the reverse mode $d = -1$ and if the photormone concentration dropped below the threshold $c^*$. if both statements are true, the magnet is disengaged, dropping any potentially picked up wall elements, and the direction parameter is set back to $d = 1$. In order to avoid the RAnt from picking up the just dropped element, it performs a random rotation in place in any direction before going back to the start of the main loop.

## Experimental set-up

The photormone was projected with an Epson EX9200 projector onto an acrylic sheet with a translucent top, which served as the surface on which the RAnts are operating. The projector uses three-chip digital light processing (DLP) which is required for the light sensors in the RAnts to pick up the photormone field. Tests with single-chip DLP projectors generated large noise in the light sensors and phototaxis was not possible. The dynamics of the photormone field is a function of the RAnt's positions and is given by

$$\partial_t c = D\nabla^2 c - k_M c + k_P \sum_{i=1}^{n} \mathcal{N}(\boldsymbol{r}_i, \boldsymbol{\Sigma}) \tag{17}$$

with $c = c(x, t)$ the photormone concentration at position $x = [x, y]$ and time $t$, $D = 10^{-5}\,\mathrm{m^2 s^{-1}}$ the diffusion coefficient, $k_M = 1\,\mathrm{s^{-1}}$ the decay rate, $k_P = 6.5\,\mathrm{s^{-1}}$ the photormone production rate, $n$ the number of RAnts detected in the arena, $\mathcal{N}(\boldsymbol{r}_i, \boldsymbol{\Sigma})$ a bivariate normal distribution with the position of the $i$ th RAnt $r_i$ as the mean and covariance $\Sigma$ with diagonal entries $\sigma^2 = 10^{-4}\mathrm{m^2}$. The position of the RAnts are used as the centers of sources of photormone. If a RAnt is not moving, photormone is built up with rate $k_P$ at that location over time and diffuses out. When the RAnt moves to a new location, the built up photormone decays with rate $k_M$. The reasoning for the parameter choices is as follows. The parameters were tuned to allow for a RAnt located at one position for one second to leave a detectable trace for 5 seconds. During that time, another RAnt moving at base speed $v_b \approx 5\mathrm{cm/s}$ can travel half the diameter of the arena. The diffusion length over the decay time scale is $\approx 3\mathrm{mm}$ which may appear small, however, RAnts are not always moving at base speed but often located in a particular location for multiple seconds to even minutes. The parameter choice described here has shown to neither saturate the domain with photormone nor be too volatile, but allowing the photormone to act as a spatiotemporal memory for the RAnts over the course of an experiment.

The positions of the RAnts are tracked with a webcam mounted above the arena and evaluated in Matlab. Blue markers are attached on the centroid of the case's upper surface which allow to use a simple blob detection to identify the pixel position of the RAnts. The photormone concentration is then dynamically updated in the same Matlab script and displayed on the RAnt arena with the projector. The tracking and integration of the photormone field is executed in real time which restricted the update rate of the projected field to $8\,Hz$ on average. The low refresh rate did not have any noticeable consequences for the conducted experiments but might have affected results for RAnts with a much larger base speed and more volatile photormone dynamics.

The set-up of the enclosure for the RAnts consisted of approximately 200 wall elements arranged in three concentric circles where the outermost circle had a diameter of $50\,cm$. The outermost circle was prevented from being pushed outward from their initial position by a thin plastic ring that was attached to the base of the arena. The plastic ring was thick enough to prevent wall elements from leaving the confinement, but thin enough for RAnts to roll over it to escape the arena. For every experiment we randomly placed the rants in the arena and waited for the first RAnt to excavate out or the time limit of 15 minutes to be reached. At that point, data was stored and the experiment ended. Most experiments required no intervention, but in case of an empty battery of a RAnt or any unexpected critical failure during the experiments, we had two RAnts standing by to replace the defective RAnt. Since all RAnts are identical and the main memory is communicated through the environment and the photormone concentration, a switch has no further statistical consequences on the outcome of the experiments. There was no leader and no dedicated roles, which makes every RAnt replaceable.

We conducted experiments for five cooperation parameters $C = \{0, 0.25, 0.5, 0.75, 1\}$ at fixed excavation rate $E = 1$ and repeated experiments five times for each parameter. Every RAnt's software was updated before a new set of five experiments with the same cooperation parameter was conducted. For every experiment, we stored the webcam data and time stamps. The video frames were post-processed and locations of all RAnts and wall elements were stored as a function of time.

For the phase diagram experiments we used the previous data for cooperation parameters $C = 0$ and $C = 1$ for partial tunneling and tunneling, respectively. To induce jamming behavior and diffusion behavior the excavation rate had to be changed in the internal programming of the RAnts. By setting the excavation rate $E = 0$ the probability of the magnet engaging vanished which led to jamming for high cooperation parameters, and diffusion for low cooperation parameters. We only collected data for two trials of a few minutes each in the diffusion and jamming case as tunneling cannot be initiated with disengaged magnets which reduces the timescales over which the behavior occurs.

## Cooperation parameter

We explored the effect of cooperation parameters on the excavation time and excavation performance as stated in the main text. Five cooperation parameters, i.e. $C = \{0, 0.25, 0.5, 0.75, 1\}$, were selected each of which was tested in five RAnt experiments with five RAnts. *Appendix 3—figure 3* shows the final wall element distribution and the RAnt density averaged over the whole trial for all the conducted 25 experiments.

From the final wall element distribution one can deduce the degree of focus during the tunneling effort. For low cooperation parameters the initial three layers are excavated at multiple excavation sites. As the cooperation parameter increases, less excavation sites are visible and at $C = 1$ there is in general only one large excavation site.

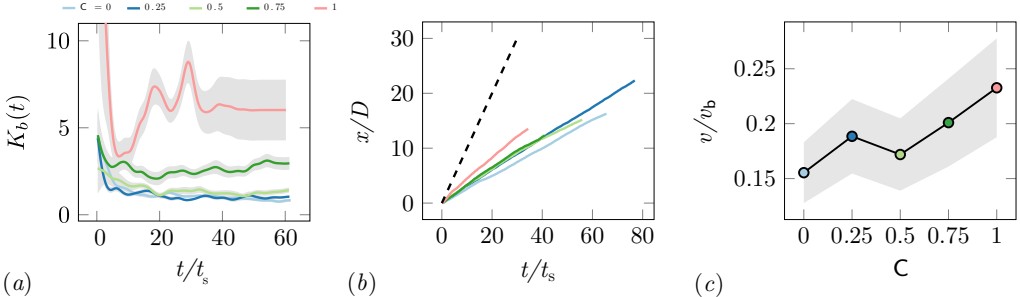

**Appendix 3—figure 2.** Characterization of the excavation using the von-Mises distribution. (**a**) Von Mises concentration parameter, $K_b$, of the angular position of the excavated boundary elements as a function of time for different cooperation parameters and averaged over 5 experiments per cooperation parameter. (**b**) Total travelled distance of RAnts for different cooperation parameters. The travelled distance $x$ is scaled by the size of the arenda $D$. The dashed line shows the travelled distance of a RAnt moving at base speed $v_b$ constantly. (**c**) RAnts' averaged speed $v$ normalized by $v_b = D/t_s$ for different cooperation parameters C.

While the final wall distribution shows only a snapshot in time, the RAnt distribution is averaged over time and therefore displays where the RAnts were mostly located throughout the run. At low cooperation numbers, the RAnt density is generally distributed all across the arena. Localization of the density toward one region was observed for low cooperation parameters as excavated wall elements were forming a new boundary that confined the RAnt motion to that region (see e.g. C = 0 T4, C = 0.25 T4). As the cooperation number increases, more distinct localized density becomes apparent. Due to the photormone field Rants operating at higher cooperation parameter values are more likely to start excavating in locations where RAnts have previously been present. The location of that attracting field is not known a priori, but emerges spontaneously through the interaction with other RAnts. The location of the peak density field at higher cooperation numbers strongly correlates with the point of excavation in the wall. The difference in RAnt behavior as a function of the cooperation parameter is the degree of focus during excavation as represented by the von Mises parameter of the angular position of excavated boundary elements shown in *Appendix 3—figure 2(a)*. A large value of the parameter indicates a high degree of concentration of the excavation effort, while low values indicate a scattered distribution of many digging sites. Another metric to assess the behavioral difference induced by the cooperation parameter is the traveled distance of the RAnts. *Appendix 3—figure 2(b)* displays the total travelled distance of a rant $x$ normalized by the arena diameter $D$ as a function of the normalized time $t/t_s$, where $t_s = D/v_b$ and $v_b$ the base speed, shows that RAnts travel a greater distance in the same amount of time at higher concentration parameters. The theoretical limit of the travelled distance is shown with the dashed line in the left-hand side figure, reflecting that RAnts do not constantly move at base speed, but are interrupted by other RAnts, obstacles, and fetching/dropping manoeuvres. As shown in *Appendix 3—figure 2(c)*, RAnts travel at about a fifth of the base speed on average. An increase of the average speed is observed as a function of the cooperation parameter, which can be explained by the fact that obstacles are more scattered at lower cooperation parameters, effectively reducing the mean free path of a RAnt.

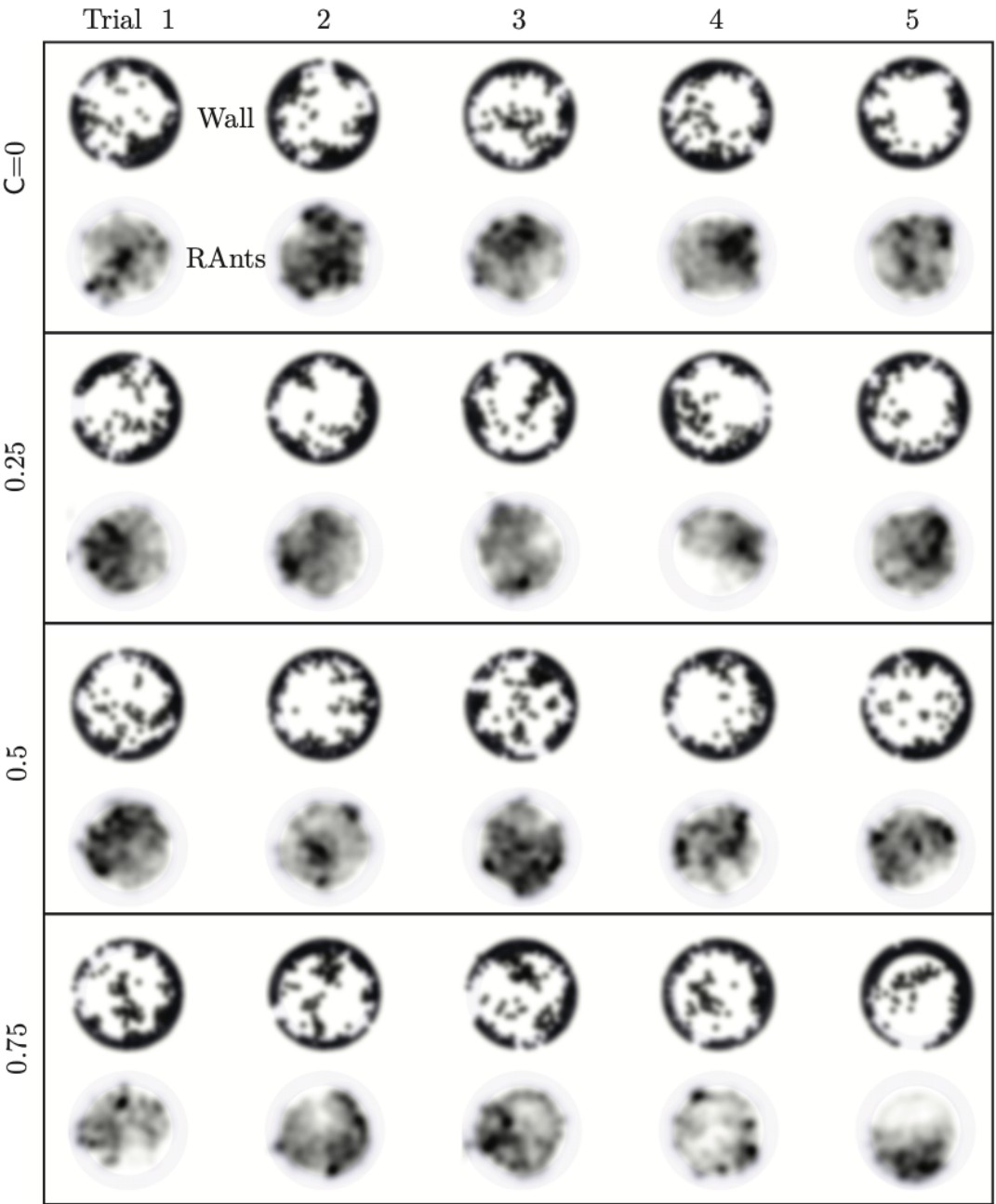

**Appendix 3—figure 3.** Final wall element distribution and averaged RAnt density field (in units of #/cm²) over the full duration of the run for all 20 experiments.

## RAnt density

In our main result we used five RAnts to explore cooperative excavation in an artificial system. More RAnts than five hindered the excavation behavior as fellow RAnts would block each others path or disturb a RAnt during the fetching and deposition of wall elements. Fewer RAnts did manage to excavate out, but the excavation rate is slower and the spontaneous formation of an excavation site due to accumulation of photormone occurs later if at all. *Appendix 3—figure 4* shows the final wall element positions and RAnt density field averaged over time for C = 1 and two experiments with one RAnt and two experiments with three RAnts.

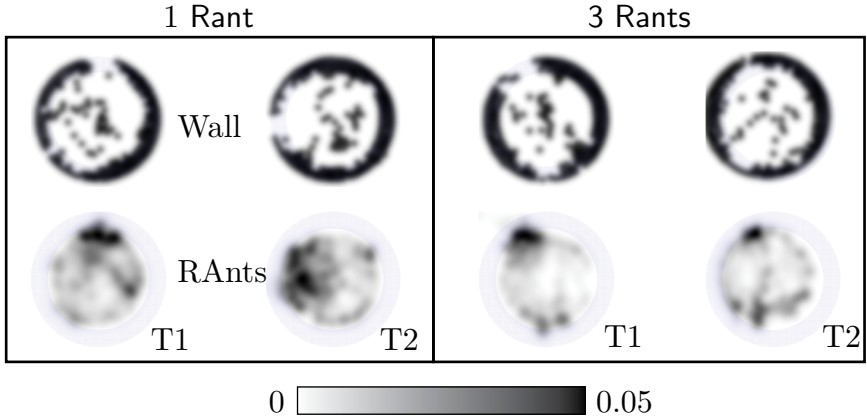

**Appendix 3—figure 4.** Final wall element distribution and averaged RAnt density field (in units of #/cm²) over the full duration of the run for experiments with one and three RAnts. The cooperation parameter was set to C = 1 and the experiments was repeated twice (Trials T1 and T2).

A single RAnt can efficiently excavate a site if an initial photormone seed is present, but it is not robust. In fact, even though the RAnt in T2 managed to remove some elements in the last layer, it never excavated out but lost the photormone seed where it was digging and started diffusing again. Three rants were more successful in generating an initial photormone seed, but excavation occured at multiple sites even for C = 1 since the lower number of RAnts did not generate one dominating photormone field.

## Phases of cooperation in RAnts

In the RAnt case we can infer the phase in which the RAnts operate by looking at the tunnel size, $1/K_b(t)$ and the location along the boundary at which the RAnts are localized, $\varrho_r(\phi)$ as they execute their task. We find that in the jammed and diffused phase there exists no tunnel and the variance remains zero throughout the process. However the location along the boundary $\phi_b$ at which the RAnts spend their time the most has a large peak around the jammed location due to high cooperation which in the case of diffusion remains widespread (see *Appendix 3—figure 5*).

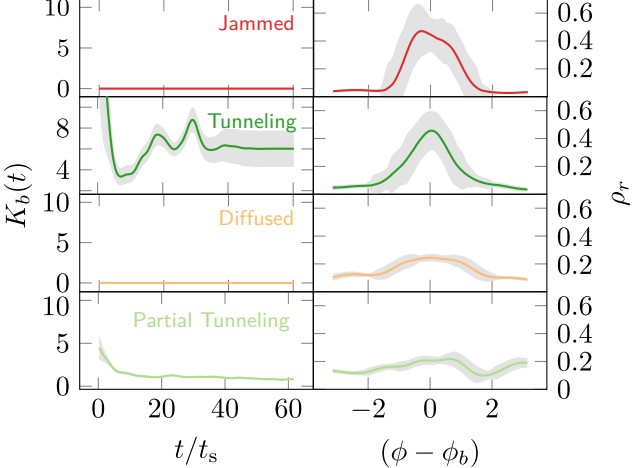

**Appendix 3—figure 5.** From the RAnt experiments in *Figure 8*, $K_b$ is the von Mises concentration parameter computed from the location of the boundary and $\varrho_r$ is the angular distribution of the RAnts in the arena averaged over time. $\phi_b$ is the time-averaged mean azimuthal location of the RAnts in the arena. RAnts present over longer periods in a particular sector of the arena will cause a peak in $\varrho_r$. One can infer the phase the RAnts are in by measuring these two quantities.

A successful tunnel, as we have already seen, has an initial increase in the variance that plateaus rapidly due to cooperation driven focus at a given location. As the RAnts are localized, focusing on

their task, we again see peaks around the location of the tunnel. For a partial tunnel, due to low cooperation, the variance in the tunnel size is large and the location along the boundary the RAnts spend their effort is spread out. Thus the phase the RAnts operate in can be distinguished by using information about the environment, i.e. the tunnel size, in combination with agent dynamics, i.e. their location.

