## [Editor Report]

This manuscript presents a quantitative study of how ants collaborate to excavate their escape from a confining barrier. The authors provide a compelling understanding of the main mechanisms driving the excavation process. They show how cooperative escape behavior arises from a non-trivial combination of movement, interaction with the substrate, and communication between individuals. The findings are supported by extensive evidence from experimental data, numerical simulations, theoretical modeling, and robotic implementation. This is an important paper that will be of interest to a broad group of researchers working on decision-making and collective behavior in living systems.

---

## [Decision Letter]

**Decision letter after peer review:**

[Editors’ note: the authors submitted for reconsideration following the decision after peer review. What follows is the decision letter after the first round of review.]

Thank you for submitting the paper "Dynamics of cooperative escape in ant and robot collectives" for consideration by *eLife*. Your article has been reviewed by 2 peer reviewers, and the evaluation has been overseen by a Reviewing Editor and a Senior Editor.

Comments to the Authors:

After consultation with the reviewers, we have decided that the paper cannot be published in the present form and that it requires major revisions.

Since *eLife* has a 60 day limit for resubmissions, and more time might be required in this case, the manuscript will be formally considered as declined at this stage. However, if you think that you can address the comments reported below and restructure the paper as suggested, you are invited to resubmit. The revised manuscript would be formally treated as a new submission but please refer to this communication and manuscript number in the cover letter. Please also submit a detailed reply to reviewers.

General comments:

Both the referees and the reviewing editor found many of the results presented in the paper potentially relevant and interesting, but they also believe that the paper should be extensively rewritten and that additional work might be required to support the conclusions.

The manuscript as it stands presents an analysis of cooperative escape using an integrated approach with experiments, theoretical modelling and a robotic implementation. Even though each of these three aspects is per se potentially interesting, the general impression is that none of them is laid out in conclusive terms and that the paper overall lacks focus.

The experimental set-up is beautifully designed, in a simple and effective manner. However, the statistics of experiments could be improved, with more experiments, and – more importantly – with a larger number of individuals. Besides, the comparison with previous experimental results and the discussion of previous literature is rather scarce (see referee's 1 report).

The theoretical model, which is in our opinion the strongest part of the paper, is presented only in a qualitative way in the main text, with all quantitative results buried in the supplementary material. Besides, a quantitative comparison with experimental data is missing.

The robotic part – even though certainly intriguing – is in itself too limited to be considered the main result of the paper, and it does not provide – in the common opinion of both referees – a solid support to the theoretical findings. Numerical simulations would have been the best way to test the coarse-grained theory and explore its main ingredients beyond the regimes where a theoretical analysis is feasible.

Our suggestion to the authors is to restructure the paper around the theoretical results on the coarse grained model, by shifting the quantitative analysis and its predictions in the main text, together with a quantitative phase diagram (see referee 2). The authors can then corroborate their theoretical predictions either by performing more experiments under varying conditions and larger sizes (referee1), or performing numerical agent-based simulations (referee 1 and 2), or – even better – both.

Such a revised manuscript would be certainly in line with the topics addressed in the Physics of Living systems section of *eLife*.

An alternative route is the one suggested by referee 1 in the report, and consists in restructuring the manuscript as a robotics paper, i.e. focusing on the robotic experiments. Even though this is certainly an interesting option, the resulting paper would however be outside the scope of *eLife* and should be submitted to a more appropriate journal.

The authors will find detailed comments in the reports attached below.*Reviewer #1 (Recommendations for the authors):*

Strengths:

– The manuscript is very well written and easy to follow, including the technical description of the model.

– The experimental system is elegant and well-suited to the goal of this research.

– The integration of experiment and theory is elegant, and the robotic platform opens up interesting, applied perspectives.

Weaknesses:

– The biological insight is limited. Excavation in ants has been fairly well studied since (at least) the 1980s (I was surprised to not see any references to classics in the field, e.g., by Tschinkel, Buhl, Rasse, Toffin, etc.) and a very similar behavioral mechanism to the one described in this study has been proposed already by Buhl in 2005 (also in different forms in 2006 and 2014). Very similar results were also found in termites (e.g., Green in 2017). In that regard, the present study does not really add anything new to our understanding of the individual and collective dynamics of excavation in social organisms, and in ants in particular.

– The functional context of the study is forced, and possibly unnatural. According to the manuscript, the function of the observed collective behavior is to escape a trap. However, a simpler explanation more in line with existing knowledge of ant behavior is that the function of this behavior is to dig tunnels in the context of underground nest construction. The escaping "function" is, here, an artifact of the structure of the experimental setup. To interpret this collective behavior as an escape behavior, the study should provide evidence that, for instance, the ants do not start excavating unless they cannot find a way out in a reasonable amount of time. Or that they would not excavate through a disk of agarose if they can move around it. This is not to say that the experimental setup is flawed (it is rather elegant to study excavation in ants) but that pitching the observed behavior as an escape behavior is grossly overinterpreting (even anthropomorphizing) the reasons behind its evolution.

– The model's predictions are fairly obvious. What the manuscript is showing is that without enough inter-attraction between the ants and a large enough excavation rate, the ants cannot focus enough activity on a single portion of the wall to get through it in a reasonable amount of time. I wish the model had been used to explore more complex questions (e.g., does the shape of the trap influence where the ants focus their digging activity? does its size/curvature impact the results?). In turn, these less obvious predictions could have been tested experimentally, either with the ants or with the robotic platform.

The robotic platform is not necessary (maybe even unproductive) for the purpose stated in the manuscript to test whether the microscopic dynamics matter. A simpler and much faster approach would have been to use an agent-based model for that purpose. This would have allowed for exploring a wider range of scenarios and parameters without losing the "synthesis" approach to animal behavior. This being said, I understand the applied appeal of the robotic implementation but then, it should have been presented as such. Robots can be useful tools to study behavior, especially when they can interact with the animals, but in this specific instance, it looks like a screwdriver was used to nail a nail. It works but there are more efficient options readily available.

– The number of experimental replicates appears to be very low. However, this may be just an impression caused by confusion in the description of the methods. The manuscript reports 7 sets of experiments, 4 with 12 major ants and 3 with a mixture of castes. However, Figure 2 seems to instead suggest that only 4 trials were conducted with 12 major ants (and presumably 3 with the mixture). If that is the case, then this study is seriously underpowered, and more trials need to be performed to make the results more robust to luck. If that is not the case, then the manuscript should be clearer regarding the number of trials performed for each experimental condition.

– First, I would recommend that the authors explore more extensively the literature on excavation behavior in ants and termites. The main biological findings are very similar to existing work done about 15 years ago and almost none of that literature is cited in the manuscript. They should then consider how to reframe their study to differentiate it from that existing knowledge.

– The manuscript should refrain from making functional claims that are not tested (or maybe even testable).

– The authors should consider extending the exploration of their model beyond making obvious predictions and they should also consider testing these predictions experimentally.

– Finally, the authors should consider reorganizing their story. At the moment, the robotic portion is a bit of a lame duck: the goal it is given here raises questions about better, more efficient methods to reach it. I think that a more compelling story would be (1) to start with the applied motivation, (2) then present the ant study as a source of inspiration for the robotic design, (3) present the robotic design and experiments, (4) extend the results from the robotic experiments with the coarse-grained model, and (5) add additional experiments to show how the model can feedback onto the robotic design. This would remove the emphasis on the biological side which does need more work to set itself apart from previous work and is also (potentially) under-powered and put the spotlight on the applied side which is, in my opinion, much stronger and more interesting.

*Reviewer #2 (Recommendations for the authors):*

The manuscript describes an interesting study of how ants collaborate to excavate their escape from a confining barrier. The experiments exhibit clear collaborative excavation, presumably induced by pheromone and antenna-driven communications between the ants. The ants aggregate along the barrier, and in these places they excavate a tunnel through the confining barrier.

A continuum model is developed, which includes the main components. This continuum model allows the authors to treat the system in a similar way to models previously developed for chemotaxis of cells, where they move up a chemical-interaction gradient that they themselves create. A simple phase diagram is charted, in terms of the strength of the coupling between the ants, and the difficulty of removing the barrier. For low coupling, the ants tend to stay diffuse, and they escape only if the barrier is soft, so that individual ants can excavate. This is a very inefficient process. At high coupling, the ants aggregate and efficiently break free if the barrier is soft enough.

Experiments using small robots explore how these components and a simple algorithm for ant behavior can capture the observations and explore regimes beyond those of the real ants. The robots interact with each other by contact, and use small magnets to grab and move aside pieces of the barrier. They communicate through a computer-generated light field that mimics the pheromone field of real ants.

The paper reports interesting work, and should be published.

1) What is the advantage gained here by using the robot-ants as opposed to a simulation on a computer? especially as the robots are not self-contained and independent agents, and require the computer-controlled light field to mimic the pheromones.

2) The continuum model has no fluctuations and noise in it. Would a simulation with self-propelled agents exhibit some behavior that is due to large fluctuations, especially for small number of ants, that is beyond the continuum model?

[Editors’ note: further revisions were suggested prior to acceptance, as described below.]

Thank you for resubmitting your work entitled "Dynamics of cooperative excavation in ant and robot collectives" for further consideration by *eLife*. Your revised article has been evaluated by Aleksandra Walczak (Senior Editor) and a Reviewing Editor.

The manuscript has been improved but there are some remaining issues that need to be addressed, as outlined below:

Even though the paper has improved with respect to the original submission, there are still several concerns expressed by the referees. Reviewer 1 (who also reviewed the original submission) is not satisfied with the authors' efforts and believes their concerns have not been addressed adequately. We, therefore, invited a new referee who had access to all previous reports and communications to express a third opinion. In their report also the third referee (referee 2 of this round) agrees that some of the original criticisms have not been addressed in the revised version. However, they believe this can be reasonably fixed.

The authors are therefore asked to very carefully address the literature comment and the question/new perspective comment. They should reply/address every point of the referee's report and try to make it clear to both referees what is the biological relevance of the work, and that an effort has been made to elucidate this aspect. We believe that this will also strengthen the work within a more biological-oriented audience.

*Reviewer #1 (Recommendations for the authors):*

Reviewer's original comment:

The biological insight is limited. Excavation in ants has been fairly well studied since (at least) the 1980s (I was surprised to not see any references to classics in the field, e.g., by Tschinkel, Buhl, Rasse, Toffin, etc.) and a very similar behavioral mechanism to the one described in this study has been proposed already by Buhl in 2005 (also in different forms in 2006 and 2014). Very similar results were also found in termites (e.g., Green in 2017). In that regard, the present study does not really add anything new to our understanding of the individual and collective dynamics of excavation in social organisms, and in ants in particular.

Authors' response:

We thank the reviewer for pointing out some of these references, we have added them to the main manuscript. We would like to highlight that the following ideas are novel to our work compared to earlier research:

– Driven by the experiments, we are able to construct a minimal model for collective excavation which points out interaction rules between the agent and the environment through the communication channel.

Reviewer's response:

This, in itself, is not new. As I mentioned above, other rather minimal models of excavation have been proposed and studied before.

Authors' response (continued):

– The theoretical model extends the classical Patlak-Keller-Segel (PKS) model for bacterial aggregation to include environmental/architectural changes. This leads to the identification of new scales and non-dimensional numbers in the system relevant to our problem, and that does not seem to have been noticed before.

– We present a unified phase-space of collective behaviors captured using the non-dimensional numbers which encapsulate and generalize the behaviors seen inside the ant nest.

Reviewer's response:

You may have indeed identified new and interesting states of that specific model, but since you never validated it against new experiments with ants, we are left with very little new biological insight. What the model is doing here is formalizing your hypothesis about the collective behavior of the ants (which is very similar to previous studies) and calculating its consequences under various conditions (i.e., making predictions). However, you never verify that these predictions hold true in the real world. For instance, you could have performed experiments with different global densities of ants to verify – at least qualitatively – your model prediction that more interaction leads to more collective tunneling activity. Similarly, you could have performed experiments with corral geometries that encourage or force either the accumulation of ants at particular locations or their even-spreading in the corral to demonstrate the same thing. Without validation experiments, the model – however thoroughly you have analyzed it – cannot be fully trusted to represent the behavior of the ants.

Reviewer's original comment:

The model's predictions are fairly obvious. What the manuscript is showing is that without enough inter-attraction between the ants and a large enough excavation rate, the ants cannot focus enough activity on a single portion of the wall to get through it in a reasonable amount of time. I wish the model had been used to explore more complex questions (e.g., does the shape of the trap influence where the ants focus their digging activity? does its size/curvature impact the results?). In turn, these less obvious predictions could have been tested experimentally, either with the ants or with the robotic platform.

Authors' response:

We are glad that the reviewer thinks our predictions are obvious after they have been pointed out and quantified.

Reviewer's response:

They are not obvious because you pointed them out. They are obvious because:

1. The excavation rate is a proxy of the work performed by each individual ant and, therefore, of the total work produced by the group. If it goes down, so does the total amount of work and, therefore, the likelihood of a successful tunnel forming.

2. The need for enough inter-attraction to generate a self-organized collective behavior is probably the most commonly made and verified prediction in the field (e.g., if ants are not sufficiently attracted toward trail pheromone, then they do not form trails or make collective decisions; if birds or fish are not sufficiently attracted towards each, then they do not form flocks or schools; etc).

Authors' response (continued):

We would like to again highlight that the contribution of our theoretical model is multi-fold: (i) the model extends the well-known PKS model to the case of ant excavation; (ii) it identifies a new non-dimensional parameter that captures how cooperation is mediated through antennation;

Reviewer's response:

On that point, why not measure the antennation rate in your experiments to validate that assumption?

Reviewer's original comment:

The robotic platform is not necessary (maybe even unproductive) for the purpose stated in the manuscript to test whether the microscopic dynamics matter. A simpler and much faster approach would have been to use an agent-based model for that purpose. This would have allowed for exploring a wider range of scenarios and parameters without losing the "synthesis" approach to animal behavior. This being said, I understand the applied appeal of the robotic implementation but then, it should have been presented as such. Robots can be useful tools to study behavior, especially when they can interact with animals, but in this specific instance, it looks like a screwdriver was used to nail a nail. It works but there are more efficient options readily available.

Authors' response:

We respectfully disagree. Building hardware and testing a system in the real world with all the "unknown unknowns" is the only real test – this we have done, in addition to simulations of agents. Furthermore, robots are not only useful because they can interact with animals; they also serve to characterize a set of sufficient rules that capture observed behavior in animals, and can gradually be pruned to understand a plausible set of necessary rules as well. We believe that our paper moves in this direction.

Reviewer's response:

A "real test" of what exactly? If the answer is "to test that the proposed behavioral algorithm produces the expected collective behavior given the robotic hardware available", then you are correct, and this is often considered the hallmark of a good swarm robotic study. However, this corresponds to an engineering question and not to a biological one. Because *eLife* is a biology journal and your stated goal in the manuscript is to answer a biology question, then the robotic portion felt out of scope in this context and better suited for a swarm robotic journal where, I believe, it would garner a lot of interest.

Now, if the answer is "to test whether the microscopic dynamics of the proposed behavioral algorithm matters" (which I believe was your original stated motivation), then it is not incorrect per se to use robots but it is very inefficient and the question could be investigated much more thoroughly with agent-based simulations, for instance, which are now included and make the robotic portion redundant.

Finally, if the answer is "to test that the proposed behavioral hypothesis is a valid representation of the observed biological collective behavior", then the robotic approach as it is described in this manuscript is not appropriate in my opinion. Indeed, robots and ants perceive stimuli, process information, and act on their environment using vastly different tools and mechanisms. In this context, showing that a given behavioral algorithm produces in robots something that resembles the collective behavior of the ants is at best showing convergence between the abilities of the two systems but not an equivalence (it would be like saying that because I can swim, I am a good model of a fish). The only "real test" in this context is done through experimental modifications of the environment of the ants or of the ants themselves (there is a long history of using drugs or performing a surgical modification to study behavior, including in social insects) in order to (in)validate specific predictions of the model.

Reviewer's original comment:

The number of experimental replicates appears to be very low. However, this may be just an impression caused by confusion in the description of the methods. The manuscript reports 7 sets of experiments, 4 with 12 major ants and 3 with a mixture of castes. However, Figure 2 seems to instead suggest that only 4 trials were conducted with 12 major ants (and presumably 3 with the mixture). If that is the case, then this study is seriously underpowered, and more trials need to be performed to make the results more robust to luck. If that is not the case, then the manuscript should be clearer regarding the number of trials performed for each experimental condition.

Authors' response:

In our experiments, we performed 7 trials with 12 ants in each trial, each lasting several hours. Given that the primary motivation of the study is to understand the mechanism of cooperative excavation, we believe increasing the number would not contribute to furthering our understanding of the mechanism.

Reviewer's response:

First, you still refer to them in the manuscript as "sets of experiments" and not individual trials. More importantly, a 7-trial experiment is well below the field's standards (e.g., in a similar experiment in Buhl et al., 2005, I count a total of 40 trials across 5 experimental conditions, each trial lasting 3 days as per the paper's methods). From what I can gather from your methods, the trials are not particularly difficult to set up, record, and automatically process, and you are not limited by the number of animals available for testing (according to AntWiki, Camponotus pennsylvanicus is abundant in the northeastern part of the US and forms large colonies). Even if one would only perform one trial per day (while, for instance, processing the data from the previous day), one would get 3 times more trials than what you have done in just under a month. Besides increasing the confidence in your observations, this could have helped you test different experimental conditions to validate the model as discussed above.

Reviewer's original comment:

First, I would recommend that the authors explore more extensively the literature on excavation behavior in ants and termites. The main biological findings are very similar to existing work done about 15 years ago and almost none of that literature is cited in the manuscript. They should then consider how to reframe their study to differentiate it from that existing knowledge.

Authors' response:

We thank the referee for insisting on improving our knowledge of and representing the existing literature. We have updated our reference list in the article and have included the relevant literature. We have also explained how our work is novel in comparison to the past.

Reviewer's response:

I'm afraid that I could not really find that discussion in the new manuscript. Where exactly do you compare your approach/model to that of previous works on the topic of excavation in social insects?

Reviewer's original comment:

Finally, the authors should consider reorganizing their story. At the moment, the robotic portion is a bit of a lame duck: the goal it is given here raises questions about better, more efficient methods to reach it. I think that a more compelling story would be (1) to start with the applied motivation, (2) then present the ant study as a source of inspiration for the robotic design, (3) present the robotic design and experiments, (4) extend the results from the robotic experiments with the coarse-grained model, and (5) add additional experiments to show how the model can feedback onto the robotic design. This would remove the emphasis on the biological side which does need more work to set itself apart from previous work and is also (potentially) under-powered and put the spotlight on the applied side which is, in my opinion, much stronger and more interesting.

Authors' response:

We thank the reviewer for the suggested reorganization, but we demur. Our goal is not to have efficient robots, but to try and quantify previous observations, create a mathematical model, and finally use both to create a robotic system that has similar properties. However, we agree that some additional work and reorganization would help, and so we have also included a new section on agent-based simulations to the existing results.

Reviewer's response:

See my previous response above about the fact that the robotic component is either inappropriate or redundant if the primary purpose of the manuscript is to investigate a biological question.

Reviewer's original comment:

L. 65-68: I am not certain how synthesizing behaviors in silico and in robots is complementary to Tin- bergen's approach. They are just a means to calculate the consequences of hypotheses made about the behavior of interest. They are modern tools to help address the four questions posed by Tinbergen, but they are not a missing part of his general approach.

Authors' response:

What we mean is that a general approach needs to be complemented by specific means to quantify behavior and synthesize it. We think that function, mechanism, development, and evolution of behavior can be complemented by synthesis that can take the form of mathematical models and physical realizations in

as much as they show what is minimally needed as a set of hypotheses framed appropriately.

Reviewer's response:

I think I understand better your point but then, I do not see how this is different from a vast majority of the work done in the field of collective behavior since the early 90s (mathematical models and robotic implementations are common in many studies since that time, and even a bit earlier) and, therefore, why it is worth emphasizing as something novel. Also, what you call synthesis is, if I understand it correctly, nothing more than using computational tools (whether embedded or not) to calculate the consequences of hypotheses about the behavior of the animals (or humans, or robots). Tinbergen's four questions were never concerned with technical approaches; they were concerned with providing a general epistemological framework for generating comprehensive explanations of animal behavior, i.e., to generate complete theories of behavior, encompassing both proximate and ultimate causes, regardless of the tools necessary to test the hypotheses underlying these theories. Proposing to "complement" them with a technical approach is missing the larger point that Tinbergen's questioning was trying to make.

Reviewer's original comment:

L. 118-119: Can you be more specific? Why exactly is that a signature of collective excavation? Even if a single ant was excavating that number would just go up as well, no?

Authors' response:

As we have explained in the Appendix, we do not observe excavation by a single ant.

Reviewer's response:

Then just say that. "Signature" is vague in this context and suggests some form of general law at work.

Reviewer's original comment:

Figure 1(d): This figure is hard to read. Why not use circular statistics to show, for instance, that the variance of the angular distribution decreases with time?

Authors' response:

We have updated SI Figure 2(f ) with the quantity suggested by the referee.

Reviewer's response:

But that doesn't help with making Figure 1(d) more readable.

*Reviewer #2 (Recommendations for the authors):*

First, I went through the manuscript, and then through the referee reports.

I found the problem – studying collective escalation in ants – of interest. And personally consider that combining experiments with ants, agent-based modeling, the coarse-grain model, and experiments with robots is a positive aspect of the study. However, the purpose of each of these different approaches to the problem is not always evident to me (e.g. agent-based and PDE models seem to provide very similar information).

One main criticism raised by the referees is that it was unclear what the objectives (or questions) of the study are. In my opinion, this has not been improved in the new version of the manuscript. Is this issue supposed to be fixed by 4 lines paragraph in red in the discussion? While I do agree that the authors identified a set of rules that reproduce the observations, it is unclear to me whether (mathematically) this is a minimal set of rules. Furthermore, I think these rules are similar to the ones used in other ant studies, including somewhere one of the authors contributed.

This is connected to another issue raised by the referees: too many relevant references were (and are) missing and there are too many self-references, whose relevance to this work is unclear (e.g. Giomi et al. Proceedings of the Royal Society A (2013)). As a consequence of this, it is difficult to identify the new aspects/contributions of the current study.

In my opinion, these issues can be easily improved.

---

## [Author Response]

[Editors’ note: the authors resubmitted a revised version of the paper for consideration. What follows is the authors’ response to the first round of review.]

General comments:Both the referees and the reviewing editor found many of the results presented in the paper potentially relevant and interesting, but they also believe that the paper should be extensively rewritten and that additional work might be required to support the conclusions.The manuscript as it stands presents an analysis of cooperative escape using an integrated approach with experiments, theoretical modelling and a robotic implementation. Even though each of these three aspects is per se potentially interesting, the general impression is that none of them is laid out in conclusive terms and that the paper overall lacks focus.

We thank the referees for their comments. We have modified large parts of the manuscript to improve focus and clarity of our work. And we have changed the title of our paper to "Dynamics of cooperative excavation in ant and robot collectives," to correctly reflect the ecologically relevant task that we study.

The experimental set-up is beautifully designed, in a simple and effective manner. However, the statistics of experiments could be improved, with more experiments, and – more importantly – with a larger number of individuals. Besides, the comparison with previous experimental results and the discussion of previous literature is rather scarce (see referee's 1 report).

We thank the editor for the positive assessment of our experimental work. As we have indicated in the Appendix, each experiment runs from 2-5 hours. During this period, we have enough statistics that allows us to first build a coarse-grained model and further confirm that there is indeed cooperation in the experiments. Besides, the reason we chose small number of individuals in the experiment was motivated by the fact that a colony *Camponotus Pennsylvanicus* often performs excavation in small groups. Further, as suggested by the referee 1, we have now included citations of previous literature extensively.

The theoretical model, which is in our opinion the strongest part of the paper, is presented only in a qualitative way in the main text, with all quantitative results buried in the supplementary material. Besides, a quantitative comparison with experimental data is missing.

We have substantially re-structured the manuscript to enhance the clarity and expanded the quantitative description of the model and its implications. We have moved important sections, especially the “Limits of phase-space” section which has the scaling analysis associated with different phases, to the main text. The lack of a one-to-one comparison between experiments and the model is because of the difficulty in measuring antennae contact statistics. This is of course a big challenge in the entire community and our future work will address these issues.

The robotic part – even though certainly intriguing – is in itself too limited to be considered the main result of the paper, and it does not provide – in the common opinion of both referees – a solid support to the theoretical findings. Numerical simulations would have been the best way to test the coarse-grained theory and explore its main ingredients beyond the regimes where a theoretical analysis is feasible.

The primary contribution of the theory is that it captures the process collective cooperative excavation by ant collectives using two non-dimensional number. We capture the inter-ant antennation via the cooperation parameter while the interaction with the environment through the excavation rate. This critical insight is what guided our robotic design and as we have detailed in the text (see Box 1), there is one-to-one correspondence between the ant system, the model and the RAnt system. In order to carry home this message further, we have now added results of an agent-based simulation to the main manuscript that is in sync with the RAnt design.

Our suggestion to the authors is to restructure the paper around the theoretical results on the coarse grained model, by shifting the quantitative analysis and its predictions in the main text, together with a quantitative phase diagram (see referee 2). The authors can then corroborate their theoretical predictions either by performing more experiments under varying conditions and larger sizes (referee1), or performing numerical agent-based simulations (referee 1 and 2), or – even better – both.Such a revised manuscript would be certainly in line with the topics addressed in the Physics of Living systems section of eLife.

We have adapted our manuscript in line with the above suggestion. The specific changes are detailed in the following responses to referees 1 and 2.

The authors will find detailed comments in the reports attached below.Reviewer #1 (Recommendations for the authors):Strengths:– The manuscript is very well written and easy to follow, including the technical description of the model.– The experimental system is elegant and well-suited to the goal of this research.– The integration of experiment and theory is elegant, and the robotic platform opens up interesting, applied perspectives.

We would like to thank the reviewer for their positive assessment of our work.

Weaknesses:– The biological insight is limited. Excavation in ants has been fairly well studied since (at least) the 1980s (I was surprised to not see any references to classics in the field, e.g., by Tschinkel, Buhl, Rasse, Toffin, etc.) and a very similar behavioral mechanism to the one described in this study has been proposed already by Buhl in 2005 (also in different forms in 2006 and 2014). Very similar results were also found in termites (e.g., Green in 2017). In that regard, the present study does not really add anything new to our understanding of the individual and collective dynamics of excavation in social organisms, and in ants in particular.

We thank the reviewer for pointing out some of these references, we have added them to the main manuscript. We would like to highlight that the following ideas are novel to our work compared to earlier research:

Driven by the experiments, we are able to construct a minimal model for collective excavation which points out interaction rules between the agent and the environment through the communication channel.The theoretical model extends the classical Patlak-Keller-Segel (PKS) model for bacterial aggregation to include environmental/architectural changes. This leads to the identification of new scales and nondimensional numbers in the system relevant for our problem, and that do not seem to have been noticed before.We present a unified phase-space of collective behaviors captured using the non-dimensional numbers which encapsulates and generalizes the behaviors seen inside the ant nest.Our robotic implementation and the simulations of the newly added agent-based models hints that beyond studying collective behavior with agents making local decisions, we can synthesize non-trivial behavior in the lab. This, we believe, is a new approach to probe the minimal (necessary and sufficient) conditions for the emergence of complex functional behavior, complementing what can be done using biological experiments.

– The functional context of the study is forced, and possibly unnatural. According to the manuscript, the function of the observed collective behavior is to escape a trap. However, a simpler explanation more in line with existing knowledge of ant behavior is that the function of this behavior is to dig tunnels in the context of underground nest construction. The escaping "function" is, here, an artifact of the structure of the experimental setup. To interpret this collective behavior as an escape behavior, the study should provide evidence that, for instance, the ants do not start excavating unless they cannot find a way out in a reasonable amount of time. Or that they would not excavate through a disk of agarose if they can move around it. This is not to say that the experimental setup is flawed (it is rather elegant to study excavation in ants) but that pitching the observed behavior as an escape behavior is grossly overinterpreting (even anthropomorphizing) the reasons behind its evolution.

We agree and have changed the functional interpretation of our data and replaced any mention of “collective escape” with “collective excavation”, and also changed the title to reflect this.

– The model's predictions are fairly obvious. What the manuscript is showing is that without enough inter-attraction between the ants and a large enough excavation rate, the ants cannot focus enough activity on a single portion of the wall to get through it in a reasonable amount of time. I wish the model had been used to explore more complex questions (e.g., does the shape of the trap influence where the ants focus their digging activity? does its size/curvature impact the results?). In turn, these less obvious predictions could have been tested experimentally, either with the ants or with the robotic platform.

We are glad that the reviewer thinks our predictions are obvious, after they have been pointed out and quantified. We would like to again highlight that the contribution of our theoretical model is multi-fold: _(*i*)_ the model extends the well known PKS model to the case of ant excavation; _(*ii*)_ it identifies a new non-dimensional parameter that captures how cooperation is mediated through antennation; _(*iii*)_ predicts a phase-space of behaviors in terms of just two parameters in the system which we then test using robotic experiments. However, as the referee points out, there are other (innumerably many) interesting questions associated with adaptability of the collective, some of which we plan to tackle in our future research.

The robotic platform is not necessary (maybe even unproductive) for the purpose stated in the manuscript to test whether the microscopic dynamics matter. A simpler and much faster approach would have been to use an agent-based model for that purpose. This would have allowed for exploring a wider range of scenarios and parameters without losing the "synthesis" approach to animal behavior. This being said, I understand the applied appeal of the robotic implementation but then, it should have been presented as such. Robots can be useful tools to study behavior, especially when they can interact with the animals, but in this specific instance, it looks like a screwdriver was used to nail a nail. It works but there are more efficient options readily available.

We respectfully disagree. Building hardware and testing a system in the real world with all the "unknown unknowns" is the only real test – this we have done, in addition to simulations of agents. Furthermore, robots are not only useful because they can interact with animals; they also serve to characterize a set of sufficient rules that capture observed behavior in animals, and can gradually be pruned to understand a plausible set of necessary rules as well. We believe that our paper moves in this direction.

– The number of experimental replicates appears to be very low. However, this may be just an impression caused by confusion in the description of the methods. The manuscript reports 7 sets of experiments, 4 with 12 major ants and 3 with a mixture of castes. However, Figure 2 seems to instead suggest that only 4 trials were conducted with 12 major ants (and presumably 3 with the mixture). If that is the case, then this study is seriously underpowered, and more trials need to be performed to make the results more robust to luck. If that is not the case, then the manuscript should be clearer regarding the number of trials performed for each experimental condition.

In our experiments we performed 7 trials with 12 ants in each trial, each lasting several hours. Given that the primary motivation of the study is to understand the mechanism of cooperative excavation, we believe increasing the number would not contribute to further our understanding of the mechanism.

– First, I would recommend that the authors explore more extensively the literature on excavation behavior in ants and termites. The main biological findings are very similar to existing work done about 15 years ago and almost none of that literature is cited in the manuscript. They should then consider how to reframe their study to differentiate it from that existing knowledge.

We thank the referee for insisting on improving our knowledge of and representing the existing literature. We have updated our reference list in the article and have included the relevant literature. We have also explained how our work is novel in comparison to the past.

– The manuscript should refrain from making functional claims that are not tested (or maybe even testable).

As suggested by the referee, we have removed all our claims of “escape” as a behavioral observation and have directed our focus on excavation, which is functional.

– The authors should consider extending the exploration of their model beyond making obvious predictions and they should also consider testing these predictions experimentally.

We have added a more thorough explanation of our model in the main manuscript, which hopefully clarifies the scope of the model (which goes beyond excavation). We have also added the simulation results of our agent-based model of the excavation process.

– Finally, the authors should consider reorganizing their story. At the moment, the robotic portion is a bit of a lame duck: the goal it is given here raises questions about better, more efficient methods to reach it. I think that a more compelling story would be (1) to start with the applied motivation, (2) then present the ant study as a source of inspiration for the robotic design, (3) present the robotic design and experiments, (4) extend the results from the robotic experiments with the coarse-grained model, and (5) add additional experiments to show how the model can feedback onto the robotic design. This would remove the emphasis on the biological side which does need more work to set itself apart from previous work and is also (potentially) under-powered and put the spotlight on the applied side which is, in my opinion, much stronger and more interesting.

We thank the reviewer for the suggested reorganization, but we demur. Our goal is not to have efficient robots, but to try and quantify previous observations, create a mathematical model and finally to use both to create a robotic system that has similar properties. However, we agree that some additional work and reorganization would help, and so we have also included a new section on agent-based simulations to the existing results.

Reviewer #2 (Recommendations for the authors):The manuscript describes an interesting study of how ants collaborate to excavate their escape from a confining barrier. The experiments exhibit clear collaborative excavation, presumably induced by pheromone and antenna-driven communications between the ants. The ants aggregate along the barrier, and in these places they excavate a tunnel through the confining barrier.A continuum model is developed, which includes the main components. This continuum model allows the authors to treat the system in a similar way to models previously developed for chemotaxis of cells, where they move up a chemical-interaction gradient that they themselves create. A simple phase diagram is charted, in terms of the strength of the coupling between the ants, and the difficulty of removing the barrier. For low coupling, the ants tend to stay diffuse, and they escape only if the barrier is soft, so that individual ants can excavate. This is a very inefficient process. At high coupling, the ants aggregate and efficiently break free if the barrier is soft enough.Experiments using small robots explore how these components and a simple algorithm for ant behavior can capture the observations and explore regimes beyond those of the real ants. The robots interact with each other by contact, and use small magnets to grab and move aside pieces of the barrier. They communicate through a computer-generated light field that mimics the pheromone field of real ants.

We thank the reviewer for the positive assessment and summary of our work.

1) What is the advantage gained here by using the robot-ants as opposed to a simulation on a computer? especially as the robots are not self-contained and independent agents, and require the computer-controlled light field to mimic the pheromones.

We have complemented the robot experiments with an agent-based simulation. Since robots are subject to similar real-world constraints as ants (friction, impacts, sensory noise, motor noise), which are often hard to model accurately, the robot experiments can address the so-called reality-gap. Even though implementing physical systems in numerical simulations can help with parameter exploration, only an embodied implementation can prove the robustness of a system to real-life constraints.

2) The continuum model has no fluctuations and noise in it. Would a simulation with self-propelled agents exhibit some behavior that is due to large fluctuations, especially for small number of ants, that is beyond the continuum model?

The continuum model averages over the fluctuations in self-propulsion as well as sensing. This is what leads to diffusive behavior captured in the density through diffusivity Da and in antennating field diffusivity Dc. To address the question of the effect of agent number on the behavior we refer to the newly added agent-based simulation in the main manuscript, where we explore this question in detail. As one would expect, we indeed find that the time of escape from the corral as a function of the agent density.

[Editors’ note: what follows is the authors’ response to the second round of review.]

The manuscript has been improved but there are some remaining issues that need to be addressed, as outlined below:Even though the paper has improved with respect to the original submission, there are still several concerns expressed by the referees. Reviewer 1 (who also reviewed the original submission) is not satisfied with the authors' efforts and believes their concerns have not been addressed adequately. We, therefore, invited a new referee who had access to all previous reports and communications to express a third opinion. In their report also the third referee (referee 2 of this round) agrees that some of the original criticisms have not been addressed in the revised version. However, they believe this can be reasonably fixed.The authors are therefore asked to very carefully address the literature comment and the question/new perspective comment. They should reply/address every point of the referee's report and try to make it clear to both referees what is the biological relevance of the work, and that an effort has been made to elucidate this aspect. We believe that this will also strengthen the work within a more biological-oriented audience.Reviewer #1 (Recommendations for the authors):Reviewer's original comment:The biological insight is limited. Excavation in ants has been fairly well studied since (at least) the 1980s (I was surprised to not see any references to classics in the field, e.g., by Tschinkel, Buhl, Rasse, Toffin, etc.) and a very similar behavioral mechanism to the one described in this study has been proposed already by Buhl in 2005 (also in different forms in 2006 and 2014). Very similar results were also found in termites (e.g., Green in 2017). In that regard, the present study does not really add anything new to our understanding of the individual and collective dynamics of excavation in social organisms, and in ants in particular.Authors' response:We thank the reviewer for pointing out some of these references, we have added them to the main manuscript. We would like to highlight that the following ideas are novel to our work compared to earlier research:– Driven by the experiments, we are able to construct a minimal model for collective excavation which points out interaction rules between the agent and the environment through the communication channel.Reviewer's response:This, in itself, is not new. As I mentioned above, other rather minimal models of excavation have been proposed and studied before.

We agree that the reference Buhl et al. 2005 is relevant to our work and have cited appropriately in the article now. However, Buhl et al. use is a 1-dimensional model which does not couple the dynamics of the 3 fields relevant to capture task execution i.e. the agent dynamics, communication/antennation field, substrate density. Moreover there is no characterization of the non-dimensional numbers in the model and the notion of a phase-space for the task of excavation which (as we have shown) determines whether a collective can successfully execute the task.

Authors' response (continued):– The theoretical model extends the classical Patlak-Keller-Segel (PKS) model for bacterial aggregation to include environmental/architectural changes. This leads to the identification of new scales and non-dimensional numbers in the system relevant to our problem, and that does not seem to have been noticed before.– We present a unified phase-space of collective behaviors captured using the non-dimensional numbers which encapsulate and generalize the behaviors seen inside the ant nest.Reviewer's response:You may have indeed identified new and interesting states of that specific model, but since you never validated it against new experiments with ants, we are left with very little new biological insight. What the model is doing here is formalizing your hypothesis about the collective behavior of the ants (which is very similar to previous studies) and calculating its consequences under various conditions (i.e., making predictions). However, you never verify that these predictions hold true in the real world. For instance, you could have performed experiments with different global densities of ants to verify – at least qualitatively – your model prediction that more interaction leads to more collective tunneling activity. Similarly, you could have performed experiments with corral geometries that encourage or force either the accumulation of ants at particular locations or their even-spreading in the corral to demonstrate the same thing. Without validation experiments, the model – however thoroughly you have analyzed it – cannot be fully trusted to represent the behavior of the ants.

As we have shown schematically in Figure 8, we do see other phases such as jammed and diffused transiently in the ant experiments. Moreover, as has been shown by Trible et al. 2017 when the antennation is muted, ants cannot cooperate and move towards a diffused phase, consistent with our predictions. We would also like to point out that the model is different from previous studies in ants where simple 1-dimensional models are used to describe the collective behavior. This is the first time that we are aware of where the 3 relevant fields i.e. agent density, antennating field and substrate density have been identified and coupled in a consistent manner, leading to emergent tasks such as excavation.

Moreover, as we have shown in Appendix 2 Figure 1, a threshold concentration for triggering excavation, and a high enough cooperation are important for successful task execution. Though the role of threshold has been separately studied in previous studies (see for example Gordon et al. 1999, Gal et al. 2022), the collective role of this threshold and the role of cooperation has not been integrated into an agent-based model or indeed a continuum model before. Our numerical simulations of the discrete and continuum model as well as the robotic experiments confirm that both these effects – the threshold for excavation and the role of cooperation are important for collective task execution.

Reviewer's original comment:The model's predictions are fairly obvious. What the manuscript is showing is that without enough inter-attraction between the ants and a large enough excavation rate, the ants cannot focus enough activity on a single portion of the wall to get through it in a reasonable amount of time. I wish the model had been used to explore more complex questions (e.g., does the shape of the trap influence where the ants focus their digging activity? does its size/curvature impact the results?). In turn, these less obvious predictions could have been tested experimentally, either with the ants or with the robotic platform.Authors' response:We are glad that the reviewer thinks our predictions are obvious after they have been pointed out and quantified.Reviewer's response:They are not obvious because you pointed them out. They are obvious because:1. The excavation rate is a proxy of the work performed by each individual ant and, therefore, of the total work produced by the group. If it goes down, so does the total amount of work and, therefore, the likelihood of a successful tunnel forming.2. The need for enough inter-attraction to generate a self-organized collective behavior is probably the most commonly made and verified prediction in the field (e.g., if ants are not sufficiently attracted toward trail pheromone, then they do not form trails or make collective decisions; if birds or fish are not sufficiently attracted towards each, then they do not form flocks or schools; etc).

We agree with the reviewer on both accounts and as we have mentioned already, we have quantified the dynamics of excavation here and have pointed out the role of simple non-dimensional numbers that characterize the amount of cooperation, excavation rate, and antennation field dynamics. The resulting model captures what we think are a minimal set of interaction rules required for successful execution of a task, as we see in our recapitulation using robotic studies. We repeat part of our answer above here as well: Though the role of threshold has been separately studied in previous studies (see for example Gordon et al. 1999, Gal et al. 2022), the collective role of this threshold and the role of cooperation has not been integrated into an agent-based model or indeed a continuum model before. Our numerical simulations of the discrete and continuum model as well as the robotic experiments confirm that both these effects – the threshold for excavation and the role of cooperation are important for collective task execution.

Authors' response (continued):We would like to again highlight that the contribution of our theoretical model is multi-fold: (i) the model extends the well-known PKS model to the case of ant excavation; (ii) it identifies a new non-dimensional parameter that captures how cooperation is mediated through antennation;Reviewer's response:On that point, why not measure the antennation rate in your experiments to validate that assumption?

We tried measuring the antennation in ant experiments but with little success due to lot of overlap in regions with high ant density resulting in failure of the tracking process. On the other hand the coarse grained representation of the ants to arrive at ant density field provides a simplified representation of the collective and further led to a model for antennation field. The information contained in antennation is complex and we assume that its dynamics and effects on the agent’s motion is analogous to pheromone concentration.

Reviewer's original comment:The robotic platform is not necessary (maybe even unproductive) for the purpose stated in the manuscript to test whether the microscopic dynamics matter. A simpler and much faster approach would have been to use an agent-based model for that purpose. This would have allowed for exploring a wider range of scenarios and parameters without losing the "synthesis" approach to animal behavior. This being said, I understand the applied appeal of the robotic implementation but then, it should have been presented as such. Robots can be useful tools to study behavior, especially when they can interact with animals, but in this specific instance, it looks like a screwdriver was used to nail a nail. It works but there are more efficient options readily available.Authors' response:We respectfully disagree. Building hardware and testing a system in the real world with all the "unknown unknowns" is the only real test – this we have done, in addition to simulations of agents. Furthermore, robots are not only useful because they can interact with animals; they also serve to characterize a set of sufficient rules that capture observed behavior in animals, and can gradually be pruned to understand a plausible set of necessary rules as well. We believe that our paper moves in this direction.Reviewer's response:A "real test" of what exactly? If the answer is "to test that the proposed behavioral algorithm produces the expected collective behavior given the robotic hardware available", then you are correct, and this is often considered the hallmark of a good swarm robotic study. However, this corresponds to an engineering question and not to a biological one. Because eLife is a biology journal and your stated goal in the manuscript is to answer a biology question, then the robotic portion felt out of scope in this context and better suited for a swarm robotic journal where, I believe, it would garner a lot of interest.Now, if the answer is "to test whether the microscopic dynamics of the proposed behavioral algorithm matters" (which I believe was your original stated motivation), then it is not incorrect per se to use robots but it is very inefficient and the question could be investigated much more thoroughly with agent-based simulations, for instance, which are now included and make the robotic portion redundant.Finally, if the answer is "to test that the proposed behavioral hypothesis is a valid representation of the observed biological collective behavior", then the robotic approach as it is described in this manuscript is not appropriate in my opinion. Indeed, robots and ants perceive stimuli, process information, and act on their environment using vastly different tools and mechanisms. In this context, showing that a given behavioral algorithm produces in robots something that resembles the collective behavior of the ants is at best showing convergence between the abilities of the two systems but not an equivalence (it would be like saying that because I can swim, I am a good model of a fish). The only "real test" in this context is done through experimental modifications of the environment of the ants or of the ants themselves (there is a long history of using drugs or performing a surgical modification to study behavior, including in social insects) in order to (in)validate specific predictions of the model.

If we can learn about biological systems by (re)creating, in a simple way, aspects of their behavior, we believe that we have learnt how to sharpen the question, and perhaps begin to answer some part of it. We have now updated the introduction of the article now to reflect the motivation for these experiments clearly, namely, that we would like to address the question of collective excavation using biological experiments, a theoretical framework, and finally recreate it using biomimetic agents. The question of whether robotic experiments belong in a biology journal is debatable, and we hope that the editors are more open about their role and potential value.

Reviewer's original comment:The number of experimental replicates appears to be very low. However, this may be just an impression caused by confusion in the description of the methods. The manuscript reports 7 sets of experiments, 4 with 12 major ants and 3 with a mixture of castes. However, Figure 2 seems to instead suggest that only 4 trials were conducted with 12 major ants (and presumably 3 with the mixture). If that is the case, then this study is seriously underpowered, and more trials need to be performed to make the results more robust to luck. If that is not the case, then the manuscript should be clearer regarding the number of trials performed for each experimental condition.Authors' response:In our experiments, we performed 7 trials with 12 ants in each trial, each lasting several hours. Given that the primary motivation of the study is to understand the mechanism of cooperative excavation, we believe increasing the number would not contribute to furthering our understanding of the mechanism.Reviewer's response:First, you still refer to them in the manuscript as "sets of experiments" and not individual trials. More importantly, a 7-trial experiment is well below the field's standards (e.g., in a similar experiment in Buhl et al., 2005, I count a total of 40 trials across 5 experimental conditions, each trial lasting 3 days as per the paper's methods). From what I can gather from your methods, the trials are not particularly difficult to set up, record, and automatically process, and you are not limited by the number of animals available for testing (according to AntWiki, Camponotus pennsylvanicus is abundant in the northeastern part of the US and forms large colonies). Even if one would only perform one trial per day (while, for instance, processing the data from the previous day), one would get 3 times more trials than what you have done in just under a month. Besides increasing the confidence in your observations, this could have helped you test different experimental conditions to validate the model as discussed above.

We did not see variability in the behavioral dynamics of ants during the excavation process across different experiments. The averaged excavation process resulted in similar localization of the agent density along the tunnel location. Therefore, we did not carry out more experiments.

Reviewer's original comment:First, I would recommend that the authors explore more extensively the literature on excavation behavior in ants and termites. The main biological findings are very similar to existing work done about 15 years ago and almost none of that literature is cited in the manuscript. They should then consider how to reframe their study to differentiate it from that existing knowledge.Authors' response:We thank the referee for insisting on improving our knowledge of and representing the existing literature. We have updated our reference list in the article and have included the relevant literature. We have also explained how our work is novel in comparison to the past.Reviewer's response:I'm afraid that I could not really find that discussion in the new manuscript. Where exactly do you compare your approach/model to that of previous works on the topic of excavation in social insects?

We have added “Our work complements and builds on earlier studies on excavation (Buhl et al. 2005, Tschinkel et al. 2004, Deneubourg et al. 1995, Deneubourg et al. 2002) in social insects that looked at the effects of population size and role of cooperation on efficiency of digging and developed 1-dimensional models to understanding the effective excavation process. We go beyond these studies by (i) quantifying the collective behavior of ants by tracking them in space-time while following the dynamics of how they interact with each other and simultaneously following the excavation of the substrate that confines them, and (ii) using our observations to develop a theoretical framework that couples the change in ant density, substrate density and the rate of antennation in space and time to capture the collective execution of the task. We also identify the non-dimensional parameters that define the range of behaviors of the agents and use this to map out the dynamics of agents in different phases using an agent-based model. (iii) We then synthesize and recreate this behavior using custom-built robots that can respond to each other and the environment to show how they can perform this collective task. An important outcome of our study is a phase diagram that shows the emergence of different collective behaviors associated with task completion as a function of just two dimensionless parameters that characterize the local rules governing the behavior of individuals and the nature of communication between agents such as ants and robots.”

Reviewer's original comment:Finally, the authors should consider reorganizing their story. At the moment, the robotic portion is a bit of a lame duck: the goal it is given here raises questions about better, more efficient methods to reach it. I think that a more compelling story would be (1) to start with the applied motivation, (2) then present the ant study as a source of inspiration for the robotic design, (3) present the robotic design and experiments, (4) extend the results from the robotic experiments with the coarse-grained model, and (5) add additional experiments to show how the model can feedback onto the robotic design. This would remove the emphasis on the biological side which does need more work to set itself apart from previous work and is also (potentially) under-powered and put the spotlight on the applied side which is, in my opinion, much stronger and more interesting.Authors' response:We thank the reviewer for the suggested reorganization, but we demur. Our goal is not to have efficient robots, but to try and quantify previous observations, create a mathematical model, and finally use both to create a robotic system that has similar properties. However, we agree that some additional work and reorganization would help, and so we have also included a new section on agent-based simulations to the existing results.Reviewer's response:See my previous response above about the fact that the robotic component is either inappropriate or redundant if the primary purpose of the manuscript is to investigate a biological question.

We have addressed this by rewriting the introduction to reflect clearly the motivation behind the robotic experiments.

We have included the paragraph: “Here we use an ecologically relevant task in carpenter ants Camponotus Pennsylvanicus: excavation and tunneling, to quantify the dynamics of successful task execution by tracking individual ants, use this to create a quantitative framework that takes the form of mathematical models for the behavior of how agents communicate and cooperate, and finally synthesize the behavior using robots that can sense and act.” “From a biological perspective, this naturally involves understanding the neural circuits, physiology and ethology of an individual. A complementary perspective at the level of the collective is that of characterizing a "crude view of the whole," which entails the quest for a small set of rules that are sufficient for task completion and the range of possible solutions that arise from these rules that might be tested experimentally. And finally, given the ability to engineer minimally responsive biomimetic agents such as robots (Rahwan et al. 2019), a question that suggests itself is that of the synthesis of effective behaviors using these agents – to explore regions of phase space that are hard to explore with social insects, before looking for them *in-vivo*, and also to learn about the robustness of these behaviors using imperfect agents in uncertain and noisy physical environments.”

Reviewer #2 (Recommendations for the authors):First, I went through the manuscript, and then through the referee reports.I found the problem – studying collective escalation in ants – of interest. And personally consider that combining experiments with ants, agent-based modeling, the coarse-grain model, and experiments with robots is a positive aspect of the study. However, the purpose of each of these different approaches to the problem is not always evident to me (e.g. agent-based and PDE models seem to provide very similar information).

We have modified the introduction to address this issue. We have added: “Here we use an ecologically relevant task in carpenter ants *Camponotus Pennsylvanicus*: excavation and tunneling, to quantify the dynamics of successful task execution by tracking individual ants, use this to create a quantitative framework that takes the form of mathematical models for the behavior of how agents communicate and cooperate, and finally synthesize the behavior using robots that can sense and act. Our work complements and builds on earlier studies on excavation (Buhl et al. 2005, Tschinkel et al. 2004, Deneubourg et al. 1995, Deneubourg et al. 2002) in social insects that looked at the effects of population size and role of cooperation on efficiency of digging and developed 1-dimensional models to understanding the effective excavation process. We go beyond these studies by (i) quantifying the collective behavior of ants by tracking them in space-time while following the dynamics of how they interact with each other and the simultaneous excavation a substrate that confines them, and (ii) use our observations to develop a theoretical framework that couples the change in ant density, substrate density and the rate of antennation in space and time to capture the collective execution of the task. We also identify the non-dimensional parameters that define the range of behaviors of the agents and use this to map out the dynamics of agents in different phases using an agent-based model. (iii) We then synthesize and recreate this behavior using custom-built robots that can respond to each other and the environment to show how they can perform this collective task. An important outcome of our study is a phase diagram that shows the emergence of different collective behaviors associated with task completion as a function of just two dimensionless parameters that characterize the local rules and the nature of communication between agents such as ants and robots.” “From a biological perspective, this naturally involves understanding the neural circuits, physiology and ethology of an individual. A complementary perspective at the level of the collective is that of characterizing a “crude view of the whole,” which entails the quest for a small set of rules that are sufficient for task completion and the range of possible solutions that arise from these rules that might be tested experimentally. And finally, given the ability to engineer minimally responsive biomimetic agents such as robots (Rahwan et al. 2019), a question that suggests itself is that of the synthesis of effective behaviors using these agents – to explore regions of phase space that are hard to explore with social insects, before looking for them *in-vivo*, and also to learn about the robustness of these behaviors using imperfect agents in uncertain and noisy physical environments.”

One main criticism raised by the referees is that it was unclear what the objectives (or questions) of the study are. In my opinion, this has not been improved in the new version of the manuscript. Is this issue supposed to be fixed by 4 lines paragraph in red in the discussion? While I do agree that the authors identified a set of rules that reproduce the observations, it is unclear to me whether (mathematically) this is a minimal set of rules. Furthermore, I think these rules are similar to the ones used in other ant studies, including somewhere one of the authors contributed.This is connected to another issue raised by the referees: too many relevant references were (and are) missing and there are too many self-references, whose relevance to this work is unclear (e.g. Giomi et al. Proceedings of the Royal Society A (2013)). As a consequence of this, it is difficult to identify the new aspects/contributions of the current study.

We agree with the reviewer and do not claim that the set of rules are minimal. However we have sufficient rules for successful excavation and minimal number of variables to capture the collective behavior i.e. agent density, environment and their interaction through the communication field. We have modified the text accordingly and added+removed the ir/relevant references from the text. We have added: “In several examples of collective task execution, a dynamic and malleable environment is often used as a communication channel, broadening the classical notion of stigmergy to include signaling via chemical, mechanical and fluidic means. But how and when individuals switch from local uncoordinated behavior to collective cooperation that translates to successful task execution in different social systems remains a relatively unexplored question. In addition to understanding the biology of functional collective behavior, a natural set of questions is also that of characterizing the theoretical underpinnings of the rules that lead to successful task completion, the range of possible solutions and finally, the synthesis of these using minimally responsive biomimetic agents such as robots.”